# GFLOWNET TRAINING BY POLICY GRADIENTS

## ABSTRACT

Generative Flow Networks (GFlowNets) have been shown with an attractive capability to generate combinatorial objects with desired properties. In this paper, we propose a policy-dependent reward that bridges the flow balance in GFlowNet training to optimizing the expected accumulated reward in traditional Reinforcement-Learning (RL). This allows us to derive policy-based GFlowNet training strategies. It is known that the training efficiency is affected by the design of backward policies in GFlowNets. We propose a coupled training strategy that can jointly solve the GFlowNet training and backward policy design. Performance analysis is provided with a theoretical guarantee of our proposed methods. We conduct experiments on both simulated and real-world datasets to verify that our policy-based strategy for GFlowNet training can outperform existing GFlowNet training strategies.

## 1 INTRODUCTION

Generative Flow Networks (GFlowNets) are a family of generative models on the space of combinatorial objects $\mathcal{X}$ (e.g. graphs composed by organizing nodes and edges in a particular manner, or strings composed of characters in a particular ordering). GFlowNets aim to solve a challenging task, sampling $x \in \mathcal{X}$ with probability proportional to some non-negative reward function $R(x)$ that defines an unnormalized distribution, where $|\mathcal{X}|$ can be enormous and the distribution modes are highly isolated by its combinatorial nature. GFlowNets (Bengio et al., 2021; 2023) decompose the process of generating or sampling $x \in \mathcal{X}$ by generating incremental trajectories that start from a null state, pass through intermediate states, and end at $x$ as the desired terminating state. These trajectory instances are interpreted as the paths along a Directed Acyclic Graph (DAG). Probability measures of trajectories are viewed as the amount of 'water' flows along the DAG, with $R(x)$ being the total flow of trajectories that end at $x$, so that following the forward generating policy defined by the measure, sampled trajectories will end at $x$ with the probability proportional to $R(x)$.

GFlowNets bear a similar form of reinforcement learning (RL) in that they both operate over Markovian Decision Processes (MDP) with a reward function $R(x)$, where nodes, edges, and node transition distributions defined by Markovian flows are considered as states, actions, and stochastic policies in MDPs. They, however, differ in the following aspects: the goal of RL problems is to learn some optimal policies that maximize the expected cumulative trajectory reward by $R$. For **value-based** RL methods, the core to achieve this is by reducing the Temporal Difference (TD) error of Bellman equations for the estimated state value function $V$ and state-action value function $Q$ (Sutton & Barto, 2018; Mnih et al., 2013). GFlowNets amortize the sampling problem into finding some Markovian flow that assigns the proper probability flow to edges (actions) so that the total flow of trajectories ending at $x$ is $R(x)$. When studying these in the lens of RL, the existing training strategies are also value-based in that they achieve the goal by keeping the balance flow equation over states of the DAG, whose difference can be measured in trajectory-wise and edge-wise ways (Bengio et al., 2021; 2023; Malkin et al., 2022a; Madan et al., 2023). Due to the similarity of GFlowNet training and RL, investigation of the relationships between them not only can deepen understanding of GFlowNets, but also helps derive better training techniques from RL. While Bengio et al. (2021) have established a relationship under the prerequisite that the forward policy implied by the flow function is uniform, this scenario is restrictive. In this work, we propose policy-dependent rewards for GFlowNet training. This bridges GFlowNets to RL in that keeping the flow balance over DAGs can be reformulated as optimizing the expected accumulated rewards in RL problems. We then derive **policy-based** training strategies for GFlowNets, which optimize the accumulated reward by its

gradients w.r.t the forward policy directly (Agarwal et al., 2019). In terms of RL, we acknowledge that the existing GFlowNet training strategies can be considered value-based and have the advantage of allowing off-policy training over policy-based methods (Malkin et al., 2022b). Value-based methods, however, face the difficulty in designing a powerful sampler that can balance the exploration and exploitation trade-off, especially when the combinatorial space is enormous with well-isolated modes. Besides, Typical annealing or random-mixing solutions with the trained forward policy may lead to the learned policy trapped in local optima. Finally, designing strategies for powerful samplers vary according to the structure and setting of modeling environments. Therefore, there is no guaranteed superiority of value-based methods over policy-based methods. The efficiency of policy-based methods depends on robust estimation of policy gradients, which can be achieved by variance reduction techniques for gradient estimation. Besides, policy-based methods allow off-policy training (Degris et al., 2012; Haarnoja et al., 2018), for example by importance sampling. Our work provides alternative ways to improve GFlowNet performance. Our contributions can be summarized as follows:

- We reformulate the GFlowNet training problem as RL over a special MDP where the reward is policy-dependent, and the underlying Markovian Chain is absorbing. We further derive policy gradients for this special MDP, and propose policy-based training strategies for GFlowNets, inspired by existing policy-based RL methods (Sutton et al., 1999; Schulman et al., 2015; Achiam et al., 2017).

- We then formulate the design of backward policies in GFlowNets also as an RL problem and propose a coupled training strategy. While finding a desired forward policy is the goal of GFlowNet training, backward policies are used as the components of training objectives. Well-designed backward policies are expected to improve training efficiency (Shen et al., 2023).

- We provide performance analyses for theoretical guarantees of our method for GFlowNet training. Our theoretical results are also accompanied by performing experiments in three application domains, hyper-grid modeling, sequence design, and Bayesian Network (BN) structure learning. The obtained experimental results serve as empirical evidence for the validity of our work and also help empirically understand the relationship between GFlowNet training and RL.

## 2 PRELIMINARIES

### 2.1 DAGS AND NOTATIONS

In a DAG $\mathcal{G} := (\mathcal{S}, \mathcal{A})$, $s \in \mathcal{S}$ denotes a state and $a \in \mathcal{A}$ denotes a directed edge $(s{\rightarrow}s')$ and $\mathcal{A} \subseteq \mathcal{S} \times \mathcal{S}$. Assuming that there is a topological ordering $\mathcal{S}_0, \ldots, \mathcal{S}_T$ for $T + 1$ disjoint subsets of $\mathcal{S}$, then $\mathcal{S} = \bigcup_{t=0}^T \mathcal{S}_t$ and an element of $\mathcal{S}_t$ is denoted as $s_t$. We use $\{\prec, \succ, \preceq, \succeq\}$ to define the partial orders between states; for example, $\forall t < t' : s_t \prec s_{t'}$. Furthermore, being acyclic means $\forall (s{\rightarrow}s') \in \mathcal{A}: s \prec s'$. For any $s \in \mathcal{S}$, we denote its parent set by $Pa_{\mathcal{G}}(s) = \{s'|(s'{\rightarrow}s) \in \mathcal{A}\}$ and its child set $Ch_{\mathcal{G}}(s) = \{s'|(s{\rightarrow}s') \in \mathcal{A}\}$. Correspondingly, We denote the edge sets start and end at $s$ as $\mathcal{A}(s) = \{(s{\rightarrow}s')|s' \in Ch_{\mathcal{G}}(s)\}$ and $\mathring{\mathcal{A}}(s) = \{(s'{\rightarrow}s)|s' \in Pa_{\mathcal{G}}(s)\}$ respectively. The complete trajectory set is defined as $\mathcal{T} = \{\tau = (s_0 \rightarrow \cdots \rightarrow s_T)|\forall(s{\rightarrow}s') \in \tau : (s{\rightarrow}s') \in \mathcal{A}\}$. We use $\tau_{\succeq s}$ to denote the sub-trajectory that starts at $s$, and $\tau_{\geq t}$ the sub-trajectory that starts at $s_t$.

For the DAG $\mathcal{G}$ in GFlowNets, we have two special states: the initial state $s^0$ with $Pa(s^0) = \emptyset$ and $S_0 = \{s^0\}$, and the final state $s^f$ with $Ch(s^f) = \emptyset$ and $S_T = \{s^f\}$. Furthermore, the terminal state set, $S_{T-1}$, covering the object set $\mathcal{X}$ with a reward function $R : \mathcal{X} \rightarrow R^+$. For notation compactness, we restrict $\mathcal{G}$ to be graded [1] so that $\forall(s{\rightarrow}s') \in \mathcal{A}, s \in \mathcal{S}_t : s' \in \mathcal{S}_{t+1}$. Accordingly, $\mathcal{A}$ can be decomposed into $\bigcup_{t=0}^{T-1} \mathcal{A}_t$ where $\mathcal{A}_t \bigcap \mathcal{A}_{t'\neq t} = \emptyset$ and $a_t \in \mathcal{A}_t$ represents an edge $(s_t{\rightarrow}s_{t+1})$ connecting $\mathcal{S}$ and $\mathcal{S}_{t+1}$. The complete trajectory set is $\mathcal{T} = \{\tau = (s_0 \rightarrow \ldots \rightarrow s_T)|s_0 = s^0, s_T = s^f; \forall t \in \{1, \ldots, T\} : (s_{t-1}{\rightarrow}s_t) \in \mathcal{A}_{t-1}\}$.

---

[1] Any DAG can be equivalently converted to be graded by adding dummy non-terminating states. Please refer to Appendix A of Malkin et al. (2022b) for more details.

## 2.2 GFLOWNETS

GFlowNets aim at efficient sampling from $P^*(x) := \frac{R(x)}{Z^*}$, where $Z^* = \sum_{x \in \mathcal{X}} R(x)$ and directly computing $Z^*$ is often challenging with typically large $|\mathcal{X}|$. To achieve this, GFlowNets define a probability measure $F(\tau) : \mathcal{T} \to \mathbb{R}^+$ (Bengio et al., 2023), termed as 'flow', so that for any event $E$, $F(E) = \sum_{\tau \in E} F(\tau)$ and the total flow $Z = F(s^0) = F(s^f)$. For any event $E$ and $E'$, $P(E) := F(E)/Z$ and $P(E|E') := \frac{F(E \cup E')}{F(E')}$. Furthermore, $F$ is restricted to be Markovian, which means that $\forall \tau = (s^0, \ldots, s_T = s^f) \in \mathcal{T}$:

$$P(\tau) = \prod_{t=1}^{T} P_F(s_t|s_{t-1}), \quad P_F(s_t|s_{t-1}) := P(s_{t-1} \to s_t | s_{t-1}) = \frac{F(s_{t-1} \to s_t)}{F(s_{t-1})}, \quad (1)$$

where $F(s \to s') = \sum_{\tau \in \{\tau | (s \to s') \in \tau\}} F(\tau)$ and $F(s) = \sum_{\tau \in \{\tau | s \in \tau\}} F(\tau)$. Similarly, we can define $P_B(s_{t-1}|s_t) := P(s_{t-1} \to s_t | s_t) = \frac{F(s_{t-1} \to s_t)}{F(s_t)}$. For a desired $F$, we require $P^{\mathsf{T}}(x) := P(x \to s_f) = P^*(x)$. As shown in Bengio et al. (2023), the necessary and sufficient condition is:

$$\sum_{s \in Pa(s')} F(s \to s') = \sum_{s'' \in Ch(s)} F(s' \to s''), \quad \forall s' \in \mathcal{S} \setminus \{s^0, s^f\}, \quad (2)$$

where we clamp $F(x \to s_f) = R(x)$ for any $x \in \mathcal{X}$.

Directly estimating the transition flow $F(s \to s')$ via the flow matching objective (Bengio et al., 2021) can suffer from the explosion of $F$ values, of which the numerical issues may lead to the failure of model training. In practice, the Trajectory Balance (TB) objective has been shown to achieve the state-of-the-art training performance (Malkin et al., 2022a). With the TB objective, the desired flow is estimated by the total flow $Z$ and a pair of forward/backward policies, $P_F(s'|s)$ and $P_B(s|s')$. The TB objective $\mathcal{L}_{TB}(P_\mathcal{D})$ of a trajectory data sampler $P_\mathcal{D}$ is defined as:

$$\mathcal{L}_{TB}(P_\mathcal{D}) := \mathbb{E}_{P_\mathcal{D}(\tau)}[L_{TB}(\tau)], \quad L_{TB}(\tau) = \left( log \frac{P_F(\tau|s_0)Z}{P_B(\tau|x)R(x)} \right)^2. \quad (3)$$

In the equation above, $P_F(\tau|s_0) = \prod_{t=1}^{T} P_F(s_t|s_{t-1})$ with $P_F(\tau) = P_F(\tau|s_0)$ and $P_F(\tau|x) = P_F(\tau)/P_F^{\mathsf{T}}(x)$ by assumptions of GFlowNets. Correspondingly, $P_B(\tau|x) = \prod_{t=1}^{T-1} P_B(s_{i-1}|s_i)$, $P_B(\tau) := (R(x)/Z^*)P_B(\tau|x)$, and $P_B(\tau|s_0) = P_B(\tau)$. Furthermore, we define $\mu(s_0 = s^0) := Z/\hat{Z}$ as the distribution over $\mathcal{S}_0$ so that $P_F(\tau) = P_{F,\mu}(\tau) := P_F(\tau|s_0)\mu(s_0)$, where $\hat{Z}$ is a constant whose value is clamped to $Z$. We define $P_{B,\rho}(\tau) := P_B(\tau|x)\rho(s_0)$ with a chosen distribution $\rho(\cdot)$.

## 3 POLICY GRADIENTS FOR GFLOWNET TRAINING

Following Malkin et al. (2022b), we first extend the relationship between the GFlowNet training strategies based on the TB objective and KL divergence. With the extended equivalence, we then introduce our policy-based and coupled training strategies for GFlowNets. Finally, we present theoretical analyses on our proposed strategies.

### 3.1 GRADIENT EQUIVALENCE

When choosing trajectories sampler $P_\mathcal{D}(\tau) = P_F(\tau)$, the gradient equivalence between using the KL divergence and TB objective has been proven (Malkin et al., 2022b). However, this forward gradient equivalence does not take the total flow estimator, $Z$, into account. Moreover, the backward gradient equivalence requires computing the expectation over $P_B^{\mathsf{T}}(x) = R(x)/Z^*$, which is not feasible. In this work, we extend the proof of the gradient equivalence to take all gradients into account and remove the dependency on unknown $Z^*$, while keeping feasible computation.

**Proposition 1.** *Given a parametrized forward policy $P_F(\cdot|\cdot;\theta)$, a backward policy $P_B(\cdot|\cdot;\phi)$, and a total flow estimator $Z(\theta)$, the gradient of the TB objective can be written as:*

$$\frac{1}{2}\nabla_\theta \mathcal{L}_{TB}(P_F;\theta) = \nabla_\theta\{D_{\mathrm{KL}}^{\mu(\cdot;\theta)}(P_F(\tau|s_0;\theta), P_B(\tau|s_0)) + \frac{1}{2}(logZ(\theta) - logZ^*)^2\}$$

$$= \nabla_\theta D_{\mathrm{KL}}^{\mu(\cdot;\theta)}(P_F(\tau|s_0;\theta), \tilde{P}_B(\tau|s_0)); \tag{4}$$

$$\frac{1}{2}\nabla_\phi \mathcal{L}_{TB}(P_{B,\rho};\phi) = \nabla_\phi D_{\mathrm{KL}}^\rho(P_B(\tau|x;\phi), P_F(\tau|x)) = \nabla_\phi D_{\mathrm{KL}}^\rho(P_B(\tau|x;\phi), \tilde{P}_F(\tau|x)). \tag{5}$$

**Remark 1.** *We note that training via TB was intrinsically done in an off-policy setting, so $\nabla\mathcal{L}_{TB}(P_\mathcal{D}) = \mathbb{E}_{P_\mathcal{D}(\tau)}[\nabla L_{TB}(\tau)]$ for any choice of $P_\mathcal{D}$.*

In the equations above, $\tilde{P}_F(\tau|x) = P_F(\tau)$ and $\tilde{P}_B(\tau|s_0) := \frac{R(x)}{\hat{Z}}P_B(\tau|x)$, denoting two unnormalized distributions of $P_F(\tau|x)$ and $P_B(\tau|s_0)$. For arbitrary distributions $p$, $q$, and $u$, $D_{\mathrm{KL}}^u(p(\cdot|s), q(\cdot|s)) := \mathbb{E}_{u(s)}[D_{\mathrm{KL}}(p(\cdot|s), q(\cdot|s))]$.

The proof is provided in Appendix A.1. As the TB objective is a special case of the Sub-Trajectory Balance (Sub-TB) objective (Madan et al., 2023), we also provide the proof of the gradient equivalence with respect to the Sub-TB objective in Appendix A.3, where the initial distribution $\mu$ becomes non-trivial.

## 3.2 RL FORMULATION OF GFLOWNET TRAINING

Inspired by the equivalence relationship in Proposition 1, we propose new reward functions that allow us to formulate GFlowNet training as RL problems, and a corresponding policy-based training strategy.

**Definition 1** (Policy-dependent Rewards). *For any action corresponding to $a := (s{\to}s') \in \mathcal{A}(s)(a \in \dot{\mathcal{A}}(s'))$, we define two reward functions as:*

$$R_F(s,a;\theta) := log\frac{\pi_F(s,a;\theta)}{\pi_B(s',a)}, \quad R_B(s',a;\phi) := log\frac{\pi_B(s',a;\phi)}{\pi_F(s,a)}, \tag{6}$$

*where $\pi_F(s,a;\theta) := P_F(s'|s;\theta), \pi_B(s',a;\phi) := P_B(s|s';\phi)$, $\pi_B(x,a)$ is clamped to $R(x)/\hat{Z}$ for $a = (x{\to}s_f)$. For any $a \notin \mathcal{A}(s)$, $R_F(s,a) := 0$. For any $a \notin \dot{\mathcal{A}}(s')$, $R_B(s',a) := 0$.*

Tuples $(\mathcal{S}, \mathcal{A}, \mathcal{G}, R_F)$ and $(\mathcal{S}, \dot{\mathcal{A}}, \mathcal{G}, R_B)$ specify two Markov Decision Processes (MDPs) with non-stationary rewards. In the MDPs, $\mathcal{G}$ specifies a deterministic transition environment such that $P(s'|s,a) = \mathbf{1}_{s':(s{\to}s')=a}$. $(\mathcal{G}, \pi_F)$ and $(\mathcal{G}, \pi_F)$ corresponds to two absorbing Markovian chains. Accordingly, the nature of DAGs allows us to define time-invariant expected value functions of states and state-action pairs, which are defined as $V_F(s) := \mathbb{E}_{P_F(\tau_{>t}|s_t)}[\sum_{t'=t}^{T-1}R_F(s_{t'},a_{t'})|s_t = s]$ and $Q_F(s,a) := \mathbb{E}_{P_F(\tau_{>t+1}|s_t,a_t)}[\sum_{t'=t}^{T-1}R_F(s_{t'},a_{t'})|s_t = s, a_t = a]$. Then we define $J_F := \mathbb{E}_{\mu(s_0)}[V_F(s_0)]$, $A_F(s,a) := Q_F(s,a) - V_F(s)$, and $d_{F,\mu}(s) := \frac{1}{T}\sum_{t=0}^{T-1}P_F(s_t = s)$. We likewise denote the functions for the backward policy as $\{V_B, Q_B, J_B, A_B, d_{B,\rho}\}$. More details are provided in Appendix B.1. By definition, $V_F(s_0) = \mathbb{E}_{P_F(\tau_{>t}|s_0)}[\sum_{t'=0}^{T-1}R_F(s_{t'},a_{t'})|s_0] = D_{\mathrm{KL}}(P_F(\tau|s_0), \tilde{P}_B(\tau|s_0))$, so $J_F = D_{\mathrm{KL}}^\mu(P_F(\tau|s_0), \tilde{P}_B(\tau|s_0))$. We can obtain $J_B = D_{\mathrm{KL}}^\rho(P_B(\tau|x), \tilde{P}_F(\tau|x))$. We can then conclude that GFlowNet training can be converted into minimizing the expected value function $J_F$ and $J_B$ by Proposition 1. By the derivation of $\nabla J_F$ and $\nabla J_B$ that are provided in Appendix B.3, minimizing $J_F$ and $J_B$ is equivalent to updating $\pi_F, \pi_B$ and $\mu$ by the following two objectives:

$$\min_{\pi_F,\mu'} T \cdot \mathbb{E}_{d_{F,\mu}(s),\pi_F'(s,a)}[A_F(s,a)] + \mathbb{E}_{\mu'(s_0)}[V_F(s_0)], \quad \min_{\pi_B'} T \cdot \mathbb{E}_{d_{B,\rho}(s),\pi_B'(s,a)}[A_B(s,a)] \tag{7}$$

where $(d_{F,\mu}, A_F, V_F)$ and $(d_{B,\rho}, A_B)$ corresponds to $(\pi_F, \mu)$ and $\pi_B$. Our policy-based method generalizes the TB-based training with $P_\mathcal{D} = P_F$ in two aspects. First, it can be shown that with gradient estimation based on a batch of trajectories, TB-based training corresponds to approximate $Q(s,a)$ empirically based on the batch data and reduces the estimation variance by a constant baseline. By contrast, $Q(s,a)$ is approximated functionally, and $V(s)$ serves as a functional baseline for

variance reduction in our policy-based method, which typically renders more robust gradient estimation Schulman et al. (2016) (see Appendix B.4). Second, our policy-based methods intrinsically correspond to minimizing the KL divergence between two distributions. This does not require $\mathcal{G}$ to be a DAG. Thus, it allows cycles for more flexible modeling of the generation process of object $x \in \mathcal{X}$, which we leave as future work.

Moreover, it is known that policy-based methods may suffer from the high variance of gradient estimators (Agarwal et al., 2019). Trust-Region Policy optimization (TRPO) (Schulman et al., 2015) is a popular policy-based method used in RL, where it is proved that it can lead to a non-decreasing performance gain via constraining the policy update size and thus increase the training stability. Then, $\pi_F$ and $\mu$ are updated by the following TRPO-based objective:

$$\min_{\pi'_F, \mu'} T \cdot \mathbb{E}_{d_{F,\mu}(s), \pi'_F(s,a)} [A_F(s,a)] + \mathbb{E}_{\mu'(s_0)}[V_F(s_0)] \quad \text{s.t. } D_{\text{KL}}^{d_{F,\mu}}(\pi_F(s,a), \pi'_F(s,a)) \leq \delta \quad (8)$$

The objective for $P_B$ can be defined in a similar way and is omitted here. Our method can be seen as a generalization of the original method to MDPs where the reward is policy-dependent, and the induced Markov chain is absorbing. We defer the corresponding proof for its performance analysis in Section 3.4. Details of model parameter updating rules by our policy-based and TRPO-based methods are provided in Appendix B.5.

### 3.3 RL FORMULATION OF GUIDED BACKWARD POLICY DESIGN

During training of GFlowNets, $(P_B, R)$ specifies the amount of desired flow that $(P_F, Z)$ is optimized to match. While $P_B(\cdot|\cdot)$ can be chosen free in principle (Bengio et al., 2023), a well-designed $P_B$ that assigns high probabilities over sub-trajectories preceding the terminating state $x$ with a high reward value $R(x)$, will improve training efficiency. Following (Shen et al., 2023), we formulate the design problem as minimizing the following objective:

$$\mathcal{L}_{TB-G}(P_B^\rho) := \mathbb{E}_{P_B^\rho(\tau)}[L_{TB-G}(\tau)], \quad L_{TB-G}(\tau; \phi) := \left( log \frac{P_B(\tau|x; \phi)}{\tilde{P}_G(\tau|x)} \right)^2 \quad (9)$$

where $P_G(\tau|x) = \prod_{t=0}^{T-1} P_G(s_{t-1}|\tau_{\geq t})$ is called the guided backward trajectory distribution, which can be non-Markovian, , and $P_G(\tau) = P_G(\tau|x)R(x)/Z^*$. As required by the training w.r.t $P_F$, objective $\mathcal{L}_{TB-G}$ aims at finding the backward policy whose Markovian flow best matches the non-Markovian flow induced by $P_G$[2].

**Proposition 2.** *Given $P_G$ and $P_B(\cdot|\cdot; \phi)$, the gradients of $\mathcal{L}_{TB-G}$ can be written as:*

$$\frac{1}{2} \nabla_\phi \mathcal{L}_{TB-G}(P_B^\rho; \phi) = \nabla_\phi D_{\text{KL}}^\rho(P_B(\tau|x; \phi), P_G(\tau|x)) \quad (10)$$

The proof can be found in Appendix A.2. Based on the proposition, we propose a new reward that allows us to formulate the backward policy design problem as an RL problem.

**Definition 2.** *Given a guided backward trajectory distribution $P_G(\tau|x)$, we define a reward function for any action $a := (s \to s') \in \dot{\mathcal{A}}(s')$ as:*

$$R_B^G(s', a; \phi) := log \frac{\pi_B(s', a; \phi)}{\pi_G(s', a)}, \quad (11)$$

*where $\pi_G(s', a) := P_G(s|\tau_{\succeq s'})$. For any $a \notin \dot{\mathcal{A}}(s')$, $R_B^G(s', a) := 0$.*

Accordingly, we denote the associated function set as $\{V_B^G, Q_B^G, J_B^G, A_B^G, d_{B,\rho}^G\}$, which are defined in a similar way as $R_B$ but replacing $P_F$ by $P_G$.

By the definition of $J_B^G$ and Proposition 2, we can conclude that $\nabla_\phi J_B^G(\phi) = \frac{1}{2} \nabla_\phi \mathcal{L}_{TB-G}(P_B^\rho; \phi)$ and the design of backward policy can be solved by minimizing $J_B^G$. The form of $P_G$ will be detailed in the experiment section for the corresponding tasks.

---

[2]By non-Markovian assumption, $P_G(\tau|x)$ can factorize in arbitrary ways condition on $x$. Here it is assumed to factorize in the backward direction for notation compactness.

**Algorithm 1** GFlowNet Training Workflow

---

**Require:** $P_F(\cdot|\cdot;\theta)$, $Z(\theta)$, $P_B(\cdot|\cdot;\phi)$, $P_G(\cdot|\cdot)$
  **for** $n = \{1, \ldots, N\}$ **do**
    $\mathcal{D} \leftarrow \{\hat{\tau}|\hat{\tau} \sim P_F(\tau;\theta)\}$
    Update $\theta$ w.r.t $R_F$ and $\mathcal{D}$
    **if** $\phi \neq \emptyset$ **then**
      $\dot{\mathcal{D}} \leftarrow \{\hat{\tau}|\forall x \in \mathcal{D} : \hat{\tau}|x \sim P_B(\tau|x)\}$
      **if** $P_G(\hat{\tau}|x) \neq P_B(\hat{\tau}|x)$ **then**
        Updated $\phi$ w.r.t $R_B^G$ and $\dot{\mathcal{D}}$
      **else**
        Updated $\phi$ w.r.t $R_B$ and $\dot{\mathcal{D}}$
      **end if**
    **end if**
  **end for**

---

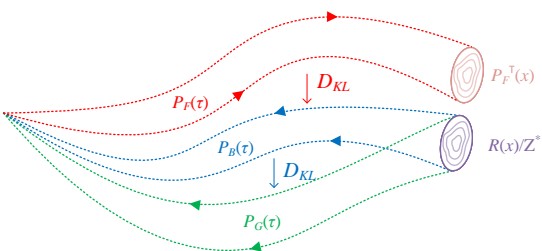

Figure 1: Dotted lines illustrate the spanning range of trajectories for one-dimensional states. $P_B$ and $P_G$ share the fixed terminating distribution over $x$. When pushing $P_F$ to match $P_B$ trajectory-wise, $P_F^\top(x)$ will also be pushed to match $R(x)/Z^*$.

Following the pipeline by Shen et al. (2023), we must solve the optimization of $\mathcal{L}_{TB-G}$ to find the desired $P_B$ at first. Then, fixing $P_B$, we can optimize $\mathcal{L}_{TB}$ to find the desired $P_F$. This gives rise to training inconvenience in practice. Since $P_G$ is not guaranteed to be a good policy, the learned $P_B$ that approximate $P_G$ may also not be a good policy for the training w.r.t $P_F$, putting the training based on $\mathcal{L}_{TB-G}$ in vain. Besides, it is troublesome if we want to dynamically design $P_B$ i.e. $P_G$ is adapted from $P_F$ currently learned. In the next section, we further show that the RL formulation allows us to optimize $J_F$ and $J_B^G$ jointly.

The workflow of our coupled training strategy is summarized in Algorithm 1 and depicted by Fig. 1.

## 3.4 PERFORMANCE ANALYSIS

In the previous sections, we formulate two RL problems with respect to $R_F$ and $R_B^G$. Now, we show below that the two problems can be solved jointly.

**Theorem 1.** *Denoting $J_F^G$ the corresponding function of $R_F$ when $P_B = P_G$ and choosing $\rho(x) = P_F^\top(x)$, $J_F^G$, $J_F$ and $J_B^G$ satisfy the following inequality:*

$$J_F^G \leq J_F + J_B^G + R_B^{G,max}\sqrt{\frac{1}{2}(J_F + logZ^* - logZ)}, \tag{12}$$

*where $R_B^{G,max} = \max_{s,a} R_B^G(s,a)$.*

The proof is given in Appendix C.1. As shown in Proposition 1, minimization of $J_F$ will incur the decrease of $D_{KL}(P_F(\tau|s_0), P_B(\tau|s_0)) = J_F + logZ^* - logZ$. Thus, by minimizing $J_F$ and $J_B^G$ jointly, the upper bound of $J_F^G$ decreases.

In the previous sections, we have also introduced the TRPO-based method besides the vanilla policy-based methods. Based on the bounds below, the TRPO-based method can be derived following a similar procedure in Schulman et al. (2015) and Achiam et al. (2017).

**Theorem 2.** *For two forward policies $(\pi_F, \pi_F')$ with $D_{KL}^{d_{F,\mu}'}(\pi_F'(\cdot, s), \pi_F(\cdot, s)) < \delta_F$, and two backward policies $(\pi_B, \pi_B')$ with $D_{KL}^{d_{B,\rho}'}(\pi_B'(\cdot, s), \pi_B(\cdot, s)) < \delta_B$*

$$
\begin{aligned}
\frac{1}{T}(J_F - J_F') &\leq E_{d_{F,\mu}'(s)\pi_F(s,a)}[A_F'(s,a)] + \epsilon_F'(2\delta_F)^{0.5} + \delta_F, \\
\frac{1}{T}(J_B - J_B') &\leq E_{d_{B,\rho}'(s)\pi_B(s,a)}[A_B'(s,a)] + \epsilon_B'(2\delta_B)^{0.5} + \delta_B,
\end{aligned}
\tag{13}
$$

*where $\epsilon_F' = max_s\mathbb{E}_{\pi_F(s,a)}[A_F'(s,a)]$ and $\epsilon_B' = max_s\mathbb{E}_{\pi_B(s,a)}[A_B'(s,a)]$. Similar results also apply to $J_B^G$ and $A_B^G$ for the backward policy $P_B$.*

The proof is given in Appendix C.2.

### 3.5 RELATED WORK

**GFlowNet training**   GFlowNets were first proposed by Bengio et al. (2021) and trained by an Flow Matching (FM) objective, which aims at minimizing the mismatch of equation 2 w.r.t a parameterized state flow estimator $F(s)$ and a parameterized edge flow estimator $F(s \rightarrow s')$ directly. Bengio et al. (2023) reformulated equation 2 and proposed a Detailed Balance (DB) objective, where edge flow $F(s \rightarrow s')$ are represented by $F(s)P_F(s'|s)$ or $F(s')P_B(s|s')$. Malkin et al. (2022a) claimed that the FM objective and DB objective are prone to inefficient credit propagation across long trajectories and showed the TB objective as the more efficient alternative. Madan et al. (2023) proposed a Sub-TB objective that unified the TB and DB objectives as special cases, which can be considered as Sub-TB objectives with sub-trajectories, which are complete or of length 1 respectively. Zimmermann et al. (2022) proposed KL-based training objectives and Malkin et al. (2022b) first established the equivalence between the KL and TB objectives. Shen et al. (2023) analyzed how the TB objective learns the desired flow under the sequence prepend/append MDP setting, and proposed a guided trajectory balance objective.

**Variational inference**   In recent approximate Bayesian inference development (Koller & Friedman, 2009), many generative models based on amortized VI (Zhang et al., 2018) and Deep Neural networks (DNNs) have shown promising potential to model complex processes (Kingma & Welling, 2014; Burda et al., 2015). These models typically involve minimizing the selected divergence measures between the target distribution and the variational distribution parametrized by DNNs. Hierarchical Variational Inference (HVI) models (Vahdat & Kautz, 2020; Zimmermann et al., 2021) generalize these models to better explore specific statistical dependency structures between observed variables and latent variables by introducing the hierarchy of latent variables. GFlowNets can be considered as a special HVI model, where non-terminating states are latent variables, the hierarchy corresponds to a DAG, and the task of minimizing divergences is achieved by keeping the flow balance (Malkin et al., 2022b). Our work provides another view of divergence minimizing by interpreting the divergence as the expected accumulated reward.

**Policy-based RL**   Policy-based RL methods typically aim to optimize the expected value function $J$ directly based on policy gradients (Sutton et al., 1999). The most relevant policy-based methods are the Actor-Critic method (Sutton & Barto, 2018) and Trust Region Policy Optimization (TRPO) (Schulman et al., 2015) along with its extension – Constrained Policy Optimization (CPO) (Achiam et al., 2017). Compared to our methods, they work under the assumption that the reward functions must be fixed w.r.t. policies. The underlying Markov chains are further assumed to be ergodic by CPO and TRPO.

**Imitation learning**   Imitation learning in RL is to learn a policy that mimics the expert demonstrations with limited expert data, by minimizing the gap between the learned policy and expert policy measured by the $0 - 1$ loss or Jensen–Shannon divergence empirically  (Rajaraman et al., 2020; Ho & Ermon, 2016). For GFlowNet training in this work, we reduce the gap between the forward policy and the expert forward policy at the trajectory level, as the expert trajectory distribution is equal to $P_B(\tau)$, implicitly encouraging the learned policy to match the desired expert policy.

**Bi-level optimization**   Our proposed training strategy can also be seen as a Stochastic Bi-level Optimization method for GFlowNet training (Ji et al., 2021; Hong et al., 2023; Ghadimi & Wang, 2018). The inner problem is the RL problem w.r.t $R_B$ or $R_B^G$ for designing backward policies. The outer problem is the RL problem w.r.t $R_F$ for forward policies. For gradient-based solutions to Bi-level optimization in general, the learning rate of inner problems is carefully selected to guarantee the overall convergence, which is not required in our method specifically designed for GFlowNet training.

## 4   EXPERIMENTS

To compare our policy-based training strategies for GFlowNets with the existing value-based strategy based on the TB objective, we have conducted three simulated experiments for hyper-grid modeling, two real-world experiments for sequence design and one study on Bayesian Network structure

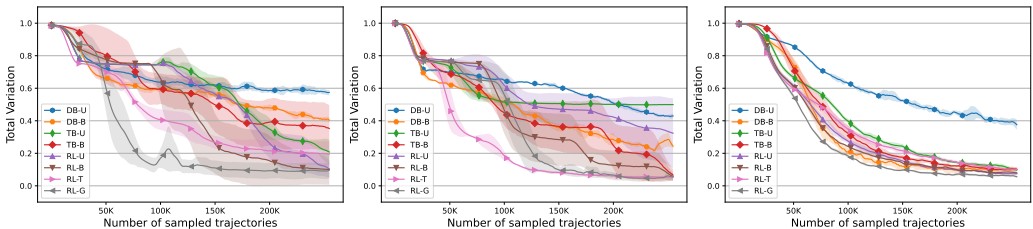

Figure 2: Training curves of $D_{TV}$ between $P_F^\top(x)$ and $P^*(x)$ for $256 \times 256$ (left), $128 \times 128 \times 128$ (middle) and $32 \times 32 \times 32 \times 32$ hyper-grids (right).

learning. The detailed descriptions of experimental settings can be found in Appendix D. We compare the performance of GFlowNets by the following training strategies: (1) DB-U: DB-based training strategy with a fixed uniform $P_B(\cdot|\cdot)$; (2) TB-U: TB-based training strategy with a fixed uniform $P_B(\cdot|\cdot)$; (3) RL-U: our proposed policy-based training strategy with a fixed uniform $P_B(\cdot|\cdot)$; (4) RL-T: our TRPO-based training strategy with a fixed uniform $P_B(\cdot|\cdot)$; (5) DB-B: DB-based training strategy with a parameterized $P_B(\cdot|\cdot)$; (6) TB-B: TB-based training strategy with a parameterized $P_B(\cdot|\cdot)$; (7) RL-B: our proposed policy-based training strategy with a parameterized $P_B(\cdot|\cdot)$; and (8) RL-G: our joint training strategy based on policy gradients with guided distribution $P_G$.

## 4.1 HYPER-GRID MODELING

In this experiment, we use the hyper-grid environment following Malkin et al. (2022b). In terms of GFlowNets, states are the coordinate tuples of an $D$-dimensional hyper-cubic grid with heights equal to $N$. The initial state $s^0$ is $\mathbf{0}$. Starting from $s^0$, actions correspond to increasing one of $D$ coordinates by 1 for the current state or stopping the process at the current state and outputting it as the terminating state $x$. A manually designed reward function $R(\cdot)$ assigns high reward values to some grid points while assigning low values to others. We conduct experiments on $256 \times 256$, $128 \times 128 \times 128$ and $32 \times 32 \times 32 \times 32$ grids. The obtained results across five runs are shown in Fig. 2 and Table 1 in the Appendix. We use the total variation $D_{TV}$ and Jensen–Shannon divergence $D_{JSD}$ to measure the gap between $P_F^\top(x)$ and $P^*(x)$, where $P_F^\top(x)$ is computed by dynamic programming. We use $T_{train}$ to denote the average training time (in minutes). The graphical illustrations of $P_F^\top(x)$ and $P^*(x)$ are shown in Figs. 7 and 8 in the Appendix.

**In the first setting**, it can be observed that our policy-based RL-U or RL-B performs better than the existing GFlowNet training methods by DB-U, TB-U, DB-B, or TB-B. With the parametrized backward policy, RL-B achieves the second-best performance. This shows that our policy-based training strategies give a more robust gradient estimation. Besides, RL-G achieves the best performance and converges much faster than all the other competing methods. In RL-G, the guided distribution assigns small values to the probability of terminating at coordinates with low rewards. This prevents the forward policy from falling into the reward 'desert' between the highly isolated modes. Finally, RL-T outperforms DB-U, TB-U, DB-B and TB-B, and it behaves more stably during training. This confirms that with the help of trust regions, the gradient estimator becomes less sensitive to environment noises. It, however, performs slightly worse than RL-U. This phenomenon may be ascribed to the choice of hyper-parameter $\delta_F$ that may over-regularize $P_F$. It is expected that using a proper scheduler of $\delta_F$ during training may further improve the performance. **In the second setting**, RL-U achieves better results than DB-U and TB-U as expected. While the performance of RL-B is similar to TB-B, it converges faster with respect to the number of sampled training trajectories. Thus, the results further support the effectiveness of our policy-based methods. Besides, RL-G and RL-T achieve the second-best and the best performances and RL-T shows faster convergence and better stability than RL-G. This again shows the superiority of coupled and TRPO-based strategies, confirming our theoretical analysis conclusions. **In the third setting**, while policy-based methods generally perform better than the flow-balance-based methods, the performance gap between policy-based methods and value-based methods is not as obvious as in the former two cases. This may be ascribed to that environment heights $H$ may have more influence on the modeling difficulty than environment dimension $N$ as hyper-grids are homogeneous w.r.t each dimension.

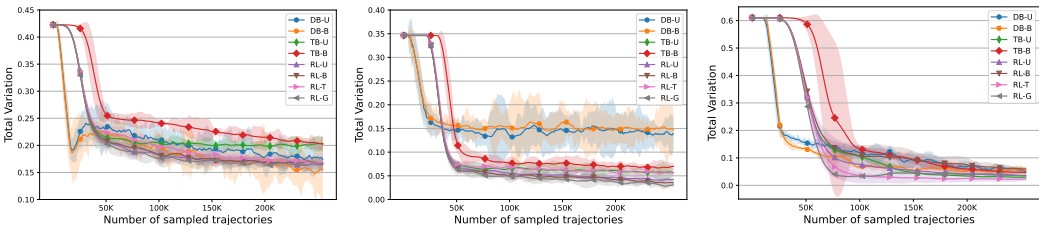

Figure 3: Training curves of $D_{TV}$ between $P_F^\top(x)$ and $P^*(x)$ for the SIX6 dataset.

Figure 4: Training curves for the QM9 dataset.

Figure 5: Training curves for the BN structure learning experiment.

## 4.2 SEQUENCE DESIGN

In this set of experiments, we use GFlowNets to generate nucleotide strings of length $D$ and molecular graphs composed of $D$ blocks according to given rewards. The initial state $s^0 = \emptyset$ is an empty sequence. The generative process runs as follows: starting from $s^0$, an action is taken to pick one of the empty slots and fill it with one element until the whole sequence is fulfilled. Then the sequence is returned as the terminating state $x$. We use the SIX6 dataset composed of strings of length 8 and the QM9 dataset composed of molecular graphs with five blocks in Shen et al. (2023). The experimental results across five runs are summarized in Table 9, Figs. 3 and 4. For the SIX6 dataset, DB-B achieves the best performance, but its stability is poor compared to the other methods. Except for DB-B, RL-G still achieves the best performance. Besides, our policy-based methods generally perform better than TB-based methods. For the QM9 dataset, our policy-based methods also perform better than TB-based methods and RL-G achieves the best performance. These results support policy-based strategies as alternative ways for GFlowNet training.

## 4.3 BAYESIAN NETWORK STRUCTURE LEARNING

In this experiment, we investigate GFlowNets for Bayesian Network (BN) structure learning following the settings adopted in Deleu et al. (2022). The set $\mathcal{X}$ in GFlowNets here corresponds to a set of BN structures, which are also DAGs. BN structure learning can be understood as approximating $P(x|\mathcal{D}) \propto R(x)$ given a dataset $\mathcal{D}$. With a set of nodes, the state space for GFlowNets is the set of all possible DAGs over the given nodes. The actions correspond to adding edges over a DAG without introducing a cycle. The generative process of a BN structure is interpreted as starting from an empty graph, an action is taken to decide to add an edge or terminate the generative process at the current graph structure. The number of possible DAGs grows exponentially with the number of nodes. We here test the same benchmark for fair comparisons with previous works (Deleu et al., 2022; Malkin et al., 2022b) with the number of nodes set to 5 and the corresponding total numbers of DAGs is about $2.92 \times 10^4$. The experimental results across five runs are shown in Fig. 5 and Table 2 in the Appendix. The graphical illustrations of $P_F^\top(x)$ and $P^*(x)$ are shown in Fig. 11. As shown in the figures, all methods achieve similar performances, which can be ascribed to the fact that the state space is relatively small. Nevertheless, RL-T achieves the best performance, and both RL-T and RL-G achieve fast convergence. These results further demonstrate the effectiveness of our policy-based methods for GFlowNet training.

## 5 CONCLUSION, LIMITATIONS AND FUTURE WORKS

This work bridges the flow-balance-based GFlowNet training to RL problems. We developed policy-based training strategies, which provide alternative ways to improve training performance compared to the existing value-based strategies. The experimental results support our claims. By formulating the training task as optimizing an expected accumulative reward, our policy-based training strategies are not limited to the cases where $\mathcal{G}$ must be a DAG. The consequent work will focus on extending the proposed methods to general $\mathcal{G}$ with the existence of cycles for more flexible modeling of generative processes. Besides, while our policy-based training strategies do not require an explicit design of a data sampler and are shown to achieve better GFlowNet training, they may still get trapped into local optima due to the variance of gradient estimation when the state space is very large. Thus. future research will also focus on further improving policy-based methods, with more robust gradient estimation according to the explained gradient equivalence relationship.

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

# A  GRADIENT EQUIVALENCE

**Lemma 1.** *(REINFORCE trick (Williams, 1992)) Given a random variable $u$ following a distribution $p(\cdot|\psi)$ parameterized by $\psi$ and a arbitrary function $f$, we have $\nabla_\psi \mathbb{E}_{p(u;\psi)}[f(u)] = \mathbb{E}_{p(u;\psi)}[f(u)\nabla_\psi log p(u;\psi)].$*

## A.1  PROOF OF PROPOSITION 1

*Proof.* First of all, we split the parameter of the total flow estimator and forward transition probability and denote them as $Z(\theta_Z)$ and $P_F(\cdot|\cdot;\theta_F)$ respectively. Besides, we define $c(\tau) = \left(log\frac{P_F(\tau|s_0)}{R(x)P_B(\tau|x)}\right)$. For the gradients w.r.t $\theta_F$:

$$
\begin{aligned}
&\frac{1}{2}\mathbb{E}_{P_F(\tau;\theta_F)}[\nabla_{\theta_F}L_{TB}(\tau;\theta_F)] = \frac{1}{2}\mathbb{E}_{P_{F,\mu}(\tau;\theta_F)}\left[\nabla_{\theta_F}\left(c(\tau;\theta_F)+logZ\right)^2\right]\\
&= \mathbb{E}_{P_{F,\mu}(\tau;\theta_F)}\left[(c(\tau;\theta_F)+logZ)\nabla_{\theta_F}logP_F(\tau|s_0;\theta_F)\right]\\
&= \mathbb{E}_{P_{F,\mu}(\tau;\theta_F)}\left[(c(\tau;\theta_F)+logZ)\nabla_{\theta_F}logP_F(\tau|s_0;\theta_F)\right] + \underbrace{\mathbb{E}_{P_{F,\mu}(\tau;\theta_F)}\left[\nabla_{\theta_F}\left(c(\tau;\theta_F)+logZ\right)\right]}_{=0}
\end{aligned}
\tag{14}
$$

where the second terms are equal to zeros as $\mathbb{E}_{P_F(\tau|s_0;\theta_F)}[\nabla_{\theta_F}c(\tau;\theta_F)] = \mathbb{E}_{P_F(\tau|s_0;\theta_F)}[\nabla_{\theta_F}logP_F(\tau|s_0;\theta_F)] = 0$ by Lemma 1. Thus,

$$
\begin{aligned}
\frac{1}{2}\mathbb{E}_{P_F(\tau;\theta_F)}[\nabla_{\theta_F}\mathcal{L}_{TB}(\tau;\theta_F)] &= \mathbb{E}_{\mu(s_0)}\left[\nabla_{\theta_F}\mathbb{E}_{P_F(\tau|s_0;\theta_F)}[c(\tau;\theta_F)+logZ]\right]\\
&= \mathbb{E}_{\mu(s_0)}\left[\nabla_{\theta_F}D_{\mathrm{KL}}(P_F(\tau|s_0;\theta_F),\tilde{P}_B(\tau|s_0))\right]\\
&= \nabla_{\theta_F}D_{\mathrm{KL}}^{\mu}(P_F(\tau|s_0;\theta_F),\tilde{P}_B(\tau|s_0))
\end{aligned}
\tag{15}
$$

Besides, we have:

$$
\begin{aligned}
&\overset{\text{e.q.15}}{=}\nabla_{\theta_F}D_{\mathrm{KL}}^{\mu}(P_F(\tau|s_0;\theta_F),\tilde{P}_B(\tau|s_0)) + \underbrace{\nabla_{\theta_F}\mathbb{E}_{P_{F,\mu}(\tau;\theta_F)}[logZ^*-logZ]}_{=\nabla_{\theta_F}(logZ^*-logZ)=0}\\
&=\nabla_{\theta_F}D_{\mathrm{KL}}^{\mu}(P_F(\tau|s_0;\theta_F),P_B(\tau|s_0))
\end{aligned}
\tag{16}
$$

The gradients w.r.t $\theta_Z$ is:

$$
\begin{aligned}
\frac{1}{2}\mathbb{E}_{P_F(\tau)}[\nabla_{\theta_Z}\mathcal{L}_{TB}(\tau;\theta_Z)] &= \frac{1}{2}\mathbb{E}_{P_F(\tau)}\left[\nabla_{\theta_Z}\left(c(\tau)+logZ(\theta_Z)\right)^2\right]\\
&= \mathbb{E}_{P_F(\tau)}\left[(c(\tau)+logZ(\theta_Z))\nabla_{\theta_Z}logZ(\theta_Z)\right]\\
&= [D_{\mathrm{KL}}(P_F(\tau|s_0),\tilde{P}_B(\tau|s_0))][\nabla_{\theta_Z}logZ(\theta_Z)]\\
&= \nabla_{\theta_Z}\frac{Z(\theta_Z)}{\hat{Z}}D_{\mathrm{KL}}(P_F(\tau|s_0),\tilde{P}_B(\tau|s_0))\\
&= \nabla_{\theta_Z}D_{\mathrm{KL}}^{\mu(\cdot;\theta_Z)}(P_F(\tau|s_0),\tilde{P}_B(\tau|s_0))
\end{aligned}
\tag{17}
$$

Besides, we have:

$$
\begin{aligned}
&\overset{\text{e.q.17}}{=}[\nabla_{\theta_Z}logZ(\theta_Z)][D_{\mathrm{KL}}(P_F(\tau|s_0;\theta_F),\tilde{P}_B(\tau|s_0))+log\frac{Z^*}{Z^*}]\\
&=[\nabla_{\theta_Z}logZ(\theta_Z)][D_{\mathrm{KL}}(P_F(\tau|s_0;\theta_F),P_B(\tau|s_0))+log\frac{Z(\theta_Z)}{Z^*}]\\
&=\nabla_{\theta_Z}\frac{Z(\theta_Z)}{\hat{Z}}[D_{\mathrm{KL}}(P_F(\tau|s_0;\theta_F),P_B(\tau|s_0))] + \left[\nabla_{\theta_Z}log\frac{Z(\theta_Z)}{Z^*}\right]\left[log\frac{Z(\theta_Z)}{Z^*}\right]\\
&=\nabla_{\theta_Z}D_{\mathrm{KL}}^{\mu(\cdot;\theta_Z)}(P_F(\tau|s_0;\theta_F),P_B(\tau|s_0)) + \frac{1}{2}\nabla_{\theta_Z}\left[log\frac{Z^{\theta_Z}}{Z^*}\right]^2
\end{aligned}
\tag{18}
$$

Combine equation 15 and 17, we obtain:

$$\frac{1}{2}\mathbb{E}_{P_F(\tau;\theta)}[\nabla_\theta L_{TB}(\tau;\theta)] = \nabla_\theta D_{\mathrm{KL}}^{\mu(\cdot;\theta)}(P_F(\tau|s_0;\theta), \tilde{P}_B(\tau|s_0))] \tag{19}$$

Combine equation 16 and 18, we obtain:

$$\frac{1}{2}\mathbb{E}_{P_F(\tau;\theta)}[\nabla_\theta L_{TB}(\tau;\theta)] = \nabla_\theta \left\{ D_{\mathrm{KL}}^{\mu(\cdot;\theta)}(P_F(\tau|s_0;\theta), P_B(\tau|s_0))] + \frac{1}{2}\left(logZ(\theta_Z) - logZ^*\right)^2 \right\} \tag{20}$$

Now let's consider the backward gradients and denote $c(\tau) = \left(log\frac{P_B(\tau|x)}{P_F(\tau)}\right)$.

$$\frac{1}{2}\mathbb{E}_{P_{B,\rho}(\tau;\phi)}[\nabla_\phi \mathcal{L}_{TB}(\tau)]$$
$$= \mathbb{E}_{P_{B,\rho}(\tau;\phi)}\left[(c(\tau;\phi) + logR(x) - logZ)\nabla_\phi logP_B(\tau|x;\phi)\right]$$
$$= \mathbb{E}_{P_{B,\rho}(\tau;\phi)}\left[c(\tau;\phi)\nabla_\phi logP_B(\tau|x;\phi)\right] + \mathbb{E}_{\rho(x)}\Big[(logR(x) - logZ)\underbrace{\mathbb{E}_{P_B(\tau|x;\phi)}[\nabla_\phi logP_B(\tau|x;\phi)]}_{=0 \text{ by Lemma 1}}\Big]$$
$$= \mathbb{E}_{P_{B,\rho}(\tau;\phi)}[c(\tau;\phi)\nabla_\phi logP_B(\tau|x;\phi)] + \underbrace{\mathbb{E}_{P_{B,\rho}(\tau;\phi)}[\nabla_\phi c(\tau;\phi)]}_{=0 \text{ by Lemma 1}}$$
$$= \mathbb{E}_{\rho(x)}[\nabla_\phi D_{\mathrm{KL}}(P_B(\tau|x;\phi), \tilde{P}_F(\tau|x))]$$
$$= \nabla_\phi D_{\mathrm{KL}}^{\rho}(P_B(\tau|x;\phi), \tilde{P}_F(\tau|x)) \tag{21}$$

Besides, we have

$$\overset{(21)}{=} \nabla_\phi D_{\mathrm{KL}}^{\rho}(P_{B,\rho}(\tau|x;\phi), \tilde{P}_F(\tau|x)) + \mathbb{E}_{\rho(x)}\Big[\underbrace{\nabla_\phi \mathbb{E}_{P_B(\tau|x;\phi)}[logP_F^\top(x)]}_{=logP_F^\top(x)\nabla_\phi 1=0}\Big]$$
$$= \nabla_\phi D_{\mathrm{KL}}^{\rho}(P_B(\tau|x;\phi), \tilde{P}_F(\tau|x)) + \nabla_\phi \mathbb{E}_{P_{B,\rho}(\tau;\phi)}[logP_F^\top(x)] \tag{22}$$
$$= \nabla_\phi D_{\mathrm{KL}}^{\rho}(P_B(\tau|x;\phi), P_F(\tau|x)) \tag{23}$$

$\square$

## A.2 PROOF OF PROPOSITION 2

*Proof.* The proof can be done by a procedure similar to that of $P_B$ in proposition 1 by replacing $\tilde{P}_F(\tau|x)$ with $P_G(\tau|x)$. $\square$

## A.3 SUB-TRAJECTORY EQUIVALENCE

Proposition 2 in the paper by Malkin et al. (2022b) only considered the gradients of the Sub-TB objective (Madan et al., 2023) w.r.t $P_F(\cdot|\cdot)$ and $P_B(\cdot|\cdot)$. We provide an extended proposition below that also takes the gradients w.r.t state flow estimator $F(\cdot)$ into consideration. For any $m < n$ and $n, m \in \{1, T-1\}$, we denote the set of sub-trajectories that start at some state in $\mathcal{S}_m$ and end in some state in $\mathcal{S}_n$ as $\bar{\mathcal{T}} = \{\bar{\tau} = (s_m \to \ldots \to s_n) | \forall i \in \{m, \ldots, n-1\} : (s_i \to s_i + 1) \in \mathcal{A}_i\}$. The sub-trajectory objective $\mathcal{L}_{Sub-TB}(P_\mathcal{D}) = \mathbb{E}_{P_\mathcal{D}(\bar{\tau})}[L_{Sub-TB}(\bar{\tau})]$ is defined by:

$$L_{Sub-TB}(\bar{\tau}) = log\left(\frac{P_F(\bar{\tau}|s_m)F(s_m)}{P_B(\bar{\tau}|s_n)F(s_n)}\right)^2, \tag{24}$$

In the equations above, $P_F(\bar{\tau}|s_m) = \prod_{t=m+1}^{n} P_F(s_t|s_{t-1})$, $P_B(\bar{\tau}|s_n) = \prod_{t=m+1}^{n} P_B(s_{t-1}|s_t)$ and $F(s_n = x)$ is clamped to $R(x)$. Besides, we define $\mu(s_m) = F(s_m)/\hat{Z}_m$ and $\rho(s_n) = F(s_n)/\hat{Z}_n$ where $\hat{Z}_m$ and $\hat{Z}_n$ are the two normalizing constants whose values are clamped to $\sum_{s_m} F(s_m)$ and $\sum_{s_n} F(s_n)$. Furthermore, we define $P_{F,\mu}(\bar{\tau}) = \mu_F(s_m)P_F(\bar{\tau}|s_m)$ and $P_B(\bar{\tau}) = \rho_B(s_n)P_B(\bar{\tau}|s_n)$ so that $P_{F,\mu}(\bar{\tau}|s_n) = P_{F,\mu}(\bar{\tau})/\rho^*(s_n)$ and $P_{B,\rho}(\bar{\tau}|s_m) = P_{B,\rho}(\bar{\tau})/\mu^*(s_m)$, where $\rho^*(s_n) = F^*(s_n)/\hat{Z}_n^*$, $F^*(s_n) = \sum_{\bar{\tau}:s_n \in \bar{\tau}} F(s_m)P_F(\bar{\tau}|s_m)$ is the ground-truth state flow over $\mathcal{S}_n$ implied by $P_F$, $\hat{Z}_n^* = \sum_{s_n} F^*(s_n)$, $\mu^*(s_m) = F^*(s_m)/\hat{Z}_m^*$, $F^*(s_m) = \sum_{\bar{\tau}:s_m \in \bar{\tau}} F(s_n)P_B(\bar{\tau}|s_n)$ is the ground-truth state flow over $\mathcal{S}_m$ implied by $P_B$, and $\hat{Z}_m^* = \sum_{s_m} F^*(s_m)$.

**Proposition 3.** *For a forward policy $P_F(\cdot|\cdot;\theta)$, a backward distribution $P_B(\cdot|\cdot;\phi)$, state flow $F(\cdot;\theta)$, and state flow $F(\cdot;\phi)^3$, the gradients of Sub-TB can be written as:*

$$\frac{1}{2}\nabla_\theta\mathcal{L}_{Sub-TB}(P_{F,\mu};\theta) = \nabla_\theta D_{\mathrm{KL}}^{\mu(\cdot;\theta)}(P_F(\bar\tau|s_m;\theta), P_{B,\rho}(\bar\tau|s_m)) + \nabla_\theta D_{\mathrm{KL}}(\mu(s_m), \mu^*(s_m))$$

$$= \nabla_\theta D_{\mathrm{KL}}^{\mu(\cdot;\theta)}(P_F(\bar\tau|s_m;\theta), \tilde{P}_{B,\rho}(\bar\tau|s_m)),$$

$$\frac{1}{2}\nabla_\phi\mathcal{L}_{Sub-TB}(P_{B,\rho};\phi) = \nabla_\phi D_{\mathrm{KL}}^{\rho(\cdot;\phi)}(P_B(\bar\tau|s_n;\phi), P_{F,\mu}(\bar\tau|s_n)) + \nabla_\theta D_{\mathrm{KL}}(\rho(s_m), \rho^*(s_m))$$

$$= \nabla_\phi D_{\mathrm{KL}}^{\rho(\cdot;\phi)}(P_B(\bar\tau|s_n;\phi), \tilde{P}_{F,\mu}(\bar\tau|s_n))$$

$$(25)$$

*where $\tilde{P}_{F,\mu}(\bar\tau|s_n) := P_{F,\mu}(\bar\tau)/\rho(s_n)$ and $\tilde{P}_{B,\rho}(\bar\tau|s_m) := P_{B,\rho}(\bar\tau)/\mu(s_m)$ are approximation to $P_{F,\mu}(\bar\tau|s_n)$ and $P_{B,\rho}(\bar\tau|s_m)$.*

*Proof.* First of all, we split the parameter of the state flow estimator and forward transition probability and denote them as $F(\cdot;\theta_M)$ and $P_F(\cdot|\cdot;\theta_F)$ respectively. Besides, we define $c(\bar\tau) = \left(log\frac{P_F(\bar\tau|s_m)}{F(s_n)P_B(\bar\tau|s_n)}\right)$

For the gradients w.r.t $\theta_F$:

$$\frac{1}{2}\mathbb{E}_{P_{F,\mu}(\bar\tau;\theta_F)}[\nabla_{\theta_F} L_{sub-TB}(\bar\tau;\theta_F)] = \frac{1}{2}\mathbb{E}_{P_{F,\mu}(\bar\tau;\theta_F)}\left[\nabla_{\theta_F}\left(c(\bar\tau;\theta_F) + logF(s_m)\right)^2\right]$$

$$= \mathbb{E}_{P_{F,\mu}(\bar\tau;\theta_F)}[(c(\bar\tau;\theta_F) + logF(s_m))\nabla_{\theta_F}logP_F(\bar\tau|s_m;\theta_F)] + \underbrace{\mathbb{E}_{P_{F,\mu}(\bar\tau;\theta_F)}[\nabla_{\theta_F}(c(\bar\tau;\theta_F) + logF(s_m))]}_{=0 \text{ by Lemma 1}}$$

$$= \mathbb{E}_{\mu(s_m)}[\nabla_{\theta_F} D_{\mathrm{KL}}(P_F(\bar\tau|s_m;\theta_F), \tilde{P}_{B,\rho}(\bar\tau|s_m)]$$

$$= \nabla_{\theta_F} D_{\mathrm{KL}}^\mu(P_F(\bar\tau|s_m;\theta_F), \tilde{P}_{B,\rho}(\bar\tau|s_m))$$

$$(26)$$

Besides,

$$\frac{1}{2}\mathbb{E}_{P_{F,\mu}(\bar\tau;\theta_F)}[\nabla_{\theta_F} L_{sub-TB}(\bar\tau;\theta_F)] = \frac{1}{2}\mathbb{E}_{P_{F,\mu}(\bar\tau;\theta_F)}\left[\nabla_{\theta_F}\left(c(\bar\tau;\theta_F) + logF(s_m)\right)^2\right]$$

$$= \mathbb{E}_{P_{F,\mu}(\bar\tau;\theta_F)}[c(\bar\tau;\theta)\nabla_{\theta_F}logP_F(\bar\tau|s_m;\theta_F)] + \mathbb{E}_{\mu(s_m)}[logF(s_m)\underbrace{\mathbb{E}_{P_F(\bar\tau|s_m;\theta_F)}[\nabla_{\theta_F}logP_F(\bar\tau|s_m;\theta_F)]}_{=0 \text{ by Lemma 1}}]$$

$$= \mathbb{E}_{P_{F,\mu}(\bar\tau;\theta_F)}[c(\bar\tau;\theta_F)\nabla_{\theta_F}logP_F(\bar\tau|s_m;\theta_F)] + \underbrace{\mathbb{E}_{P_{F,\mu}(\bar\tau;\theta_F)}[\nabla_{\theta_F}c(\bar\tau;\theta_F)]}_{=0}$$

$$= \nabla_{\theta_F}\mathbb{E}_{P_{F,\mu}(\bar\tau;\theta_F)}[c(\bar\tau;\theta_F)] + \mathbb{E}_{\mu(s_m)}[\underbrace{\nabla_{\theta_F}\mathbb{E}_{P_F(\bar\tau|s_m;\theta_F)}[log\mu^*(s_m)]}_{=log\mu^*(s_m)\nabla_{\theta_F}1=0}] + \nabla_{\theta_F}log\hat{Z}_n$$

$$= \nabla_{\theta_F}\mathbb{E}_{P_{F,\mu}(\bar\tau;\theta_F)}[c(\bar\tau;\theta_F)] + \nabla_{\theta_F}\mathbb{E}_{P_{F,\mu}(\bar\tau;\theta_F)}[log\mu^*(s_m) + log\hat{Z}_n]$$

$$= \nabla_{\theta_F} D_{\mathrm{KL}}^\mu(P_F(\bar\tau|s_m;\theta_F), P_{B,\rho}(\bar\tau|s_m))]$$

$$(27)$$

For the gradients w.r.t $\theta_M$, we have:

$$\frac{1}{2}\mathbb{E}_{P_{F,\mu}(\bar\tau;\theta_M)}[\nabla_{\theta_M} L_{sub-TB}(\bar\tau;\theta_M)] = \frac{1}{2}\mathbb{E}_{P_{F,\mu}(\bar\tau;\theta_M)}\left[\nabla_{\theta_M}\left(c(\bar\tau) + logF(s_m;\theta_M)\right)^2\right]$$

$$= \mathbb{E}_{P_{F,\mu}(\bar\tau;\theta_M)}[(c(\bar\tau) + logF(s_m;\theta_M))\nabla_{\theta_F}logF(s_m;\theta_M)]$$

$$= \mathbb{E}_{\mu(s_m;\theta_M)}\left[\nabla_{\theta_F}log\frac{F(s_m;\theta_M)}{\hat{Z}_m} D_{\mathrm{KL}}(P_F(\bar\tau|s_m;\theta_M), \tilde{P}_B(\bar\tau|s_m))\right]$$

$$= \nabla_{\theta_M} D_{\mathrm{KL}}^{\mu(\cdot;\theta_M)}(P_F(\bar\tau|s_m), \tilde{P}_B(\bar\tau|s_m))$$

$$(28)$$

---

[3]Here the $F_\theta$ and $F_\phi$ actually share the same parameters and represent the same flow estimator $F$. Model parameters are duplicated just for the clearness of gradient equivalences. Therefore the true gradient of the state flow estimator $F$ is $\nabla_\theta F_\theta + \nabla_\phi F_\phi$.

Besides,

$$\frac{1}{2}\mathbb{E}_{P_{F,\mu}(\bar{\tau};\theta_M)}[\nabla_{\theta_M}L_{sub-TB}(\bar{\tau};\theta_M)] = \frac{1}{2}\mathbb{E}_{P_{F,\mu}(\bar{\tau};\theta_M)}\left[\nabla_{\theta_M}(c(\bar{\tau})+logF(s_m;\theta_M))^2\right]$$

$$= \mathbb{E}_{P_{F,\mu}(\bar{\tau};\theta_M)}[c(\bar{\tau})\nabla_{\theta_F}logF(s_m;\theta_M)] + \mathbb{E}_{\mu(s_m;\theta_M)}[logF(s_m;\theta_M)\nabla_{\theta_M}logF(s_m;\theta_M)]$$

$$= \mathbb{E}_{P_{F,\mu}(\bar{\tau};\theta_M)}\left[(c(\bar{\tau})+log\mu^*(s_m)+log\hat{Z}_n)\nabla_{\theta_M}logF(s_m;\theta_M)\right] +$$

$$\mathbb{E}_{\mu(s_m;\theta_M)}\left[(logF(s_m;\theta_M)-log\mu^*(s_m)-log\hat{Z}_m)\nabla_{\theta_M}logF(s_m;\theta_M)\right] \quad (29)$$

where $\mathbb{E}_{P_{F,\mu}(\bar{\tau};\theta_M)}\left[log\hat{Z}_n\nabla_{\theta_M}logF(s_m;\theta_M)\right] = \mathbb{E}_{P_{F,\mu}(\bar{\tau};\theta_M)}\left[log\hat{Z}_m\nabla_{\theta_M}logF(s_m;\theta_M)\right] = 0$ by Lemma 1.

$$\stackrel{\text{e.q 29}}{=} \mathbb{E}_{\mu(s_m;\theta_M)}\left[\nabla_{\theta_M}log\frac{F(s_m;\theta_M)}{\hat{Z}_m}D_{\text{KL}}(P_F(\bar{\tau}|s_m),P_{B,\rho}(\bar{\tau}|s_m))\right] +$$

$$\mathbb{E}_{\mu(s_m;\theta_M)}\left[\left(log\frac{F(s_m;\theta_M)}{\hat{Z}_m}-log\mu^*(s_m)\right)\nabla_{\theta_M}log\frac{F(s_m;\theta_M)}{\hat{Z}_m}\right]$$

$$= \nabla_{\theta_M}D_{\text{KL}}^{\mu(\cdot;\theta_M)}(P_F(\bar{\tau}|s_m),P_{B,\rho}(\bar{\tau}|s_m)) + \nabla_{\theta_M}D_{\text{KL}}(\mu(s_m;\theta_M),\mu^*(s_m)) \quad (30)$$

Combining equation 26 and 28, we obtain

$$\frac{1}{2}\mathbb{E}_{P_{F,\mu}(\bar{\tau};\theta)}[\nabla_{\theta}L_{sub-TB}(\bar{\tau};\theta)] = \nabla_{\theta}D_{\text{KL}}^{\mu(\cdot;\theta)}(P_F(\bar{\tau}|s_m;\theta),\tilde{P}_B(\bar{\tau}|s_m)) \quad (31)$$

Combining equation 27 and 30, we obtain

$$\frac{1}{2}\mathbb{E}_{P_{F,\mu}(\bar{\tau};\theta)}[\nabla_{\theta}L_{sub-TB}(\bar{\tau};\theta)] = \nabla_{\theta}\{D_{\text{KL}}^{\mu(\cdot;\theta)}(P_F(\bar{\tau}|s_m;\theta),P_B(\bar{\tau}|s_m))+D_{\text{KL}}(\mu(s_m;\theta),\mu^*(s_m))\}$$
$$(32)$$

Splitting $\phi$ into $\phi_B$ and $\phi_M$ and denoting $c(\bar{\tau}) = log\frac{P_B(\bar{\tau}|s_n)}{F(s_m)P_F(\bar{\tau}|s_m)}$, the gradient derivations of $\phi$ follows the similar way as $\theta$. $\qquad\square$

# B  RL FRAMEWORK

## B.1  DERIVATION OF RL FUNCTIONS

Let's first consider the case of forward policies. For any $s \in \mathcal{S}_t$ and $a = (s{\to}s') \in \mathcal{A}(s)$ with $t \in \{0,\ldots,T-1\}$, we define the $V_{F,t}$ and $Q_{F,t}$ as:

$$V_{F,t}(s) := \mathbb{E}_{P_F(\tau_{>t}|s_t)}\left[\sum_{t'=t}^{T-1}R_F(s_{t'},a_{t'})\bigg|s_t=s\right]$$

$$= R_F(s) + \mathbb{E}_{P_F(s_{t+1}|s_t)}\left[\mathbb{E}_{P_F(\tau_{>t+1}|s_{t+1})}\left[\sum_{t'=t+1}^{T}R_F(s_{t'},a_{t'})\bigg|s_{t+1}=s'\right]\bigg|s_t=s\right]$$

$$= R_F(s) + \mathbb{E}_{\pi_F(s,a)}[V_{F,t+1}(s')]$$

$$Q_{F,t}(s,a) := \mathbb{E}_{P_F(\tau_{>t+1}|s_t,a_t)}\left[\sum_{t'=t}^{T-1}R_F(s_{t'},a_{t'})\bigg|s_t=s,a_t=a\right]$$

$$= R_F(s,a) + \mathbb{E}_{P_F(\tau_{>t+1}|s_{t+1})}\left[\sum_{t'=t+1}^{T}R_F(s_{t'},a_{t'})\bigg|s_{t+1}=s'\right]$$

$$= R_F(s,a) + V_{F,t+1}(s')$$
$$(33)$$

where $R_F(s) := \mathbb{E}_{\pi_F(s,a)}[R_F(s,a)]$, $V_{F,T}(\cdot) := 0$, and $Q_{F,T}(\cdot,\cdot) := 0$. Since $S_t \cap S_{t'} = \emptyset$ for any $t \neq t'$, we can read off the time indices (topological orders) from state values. Plus the fact that $R_F(s,a) := 0$ for any $a \notin A(s)$, we are allowed to define two universal $V_F : \mathcal{S} \to \mathbb{R}$ and $Q_F : \mathcal{S} \times \mathcal{A} \to \mathbb{R}$ such that $V_F(s_t=s) := V_{F,t}(s)$ and $Q_F(s_t=s,a) := Q_{F,t}(s,a)$.

**Remark 2.** *While the transition environment $\mathcal{G}$ is exactly known, the state space $\mathcal{S}$ can be exponentially large, so the exact value of $V$ and $Q$ is intractable. This, in spirit, corresponds to a regular RL problem where the exact values of $V$ and $Q$ are infeasible due to the unknown and uncertain transition model $P(s'|s, a)$.*

For backward policies, rewards are accumulated from time $T$ to 1. Similarly, for $s' \in \mathcal{S}_t$ and $a = (s \rightarrow s') \in \dot{\mathcal{A}}(s')$ we can define:

$$
V_{B,t}(s') := \mathbb{E}_{P_B(\tau_{<t}|s_t)} \left[ \sum_{t'=1}^{t} R_B(s_{t'}, a_{t'}) \middle| s_t = s' \right] = R_B(s') + \mathbb{E}_{\pi_B(s',a)}[V_{B,t-1}(s)]
$$

$$
Q_{B,t}(s', a) := \mathbb{E}_{P_B(\tau_{<t-1}|s_t,a_t)} \left[ \sum_{t'=1}^{t} R_B(s_{t'}, a_{t'}) \middle| s_t = s', a_t = a \right] = R_B(s', a) + V_{B,t-1}(s)
$$

$$(34)$$

where $R_B(s') = \mathbb{E}_{\pi_B(s',a)}[R_B(s', a)]$, $V_{B,0}(\cdot) := 0$, and $Q_{B,0}(\cdot, \cdot) := 0$. For the same reason as forward policies, we can define universal functions $V_B : \mathcal{S} \rightarrow R$ and $Q_B : \mathcal{S} \times \dot{\mathcal{A}} \rightarrow R$ such that $V_B(s_t = s') := V_{B,t}(s')$ and $Q_B(s_t = s', a) := Q_{B,t}(s', a)$.

The expected value functions are defined as:

$$
J_F = \mathbb{E}_{\mu(s_0)}[V_F(s_0)] = D_{\mathrm{KL}}^{\mu}(P_F(\tau|s_0), \tilde{P}_B(\tau|s_0)
$$
$$
J_B = \mathbb{E}_{\rho(x)}[V_B(x)] = D_{\mathrm{KL}}^{\rho}(P_B(\tau|x), \tilde{P}_F(\tau|x)).
$$

$$(35)$$

The advantages functions are defined as: $A_F(s, a) = Q_F(s, a) - V_F(s)$ and $A_B(s, a) = Q_B(s, a) - V_B(s)$.

We define forward accumulated state distribution as $d_{F,\mu}(s) := \frac{1}{T} \sum_{t=0}^{T-1} P_{F,\mu}(s_t)$ such that for arbitrary function $f : \mathcal{S} \times \mathcal{A} \rightarrow \mathbb{R}$,

$$
\mathbb{E}_{P_{F,\mu}(\tau)} \left[ \sum_{t=0}^{T-1} f(s_t, a_t) \right] = \sum_{t=0}^{T} \mathbb{E}_{P_{F,\mu}(s_t, s_{t+1})}[f(s_t, a_t)] = \sum_{t=0}^{T} \mathbb{E}_{P_{F,\mu}(s_t), \pi(s_t, a_t)}[f(s_t, a_t)]
$$

$$
= \sum_{t}^{T} \sum_{s}^{\mathcal{S}} P_{F,\mu}(s_t = s) \sum_{a}^{\mathcal{A}} \pi(s, a) f(s, a) \qquad (36)
$$

$$
= \sum_{s}^{\mathcal{S}} \sum_{a}^{\mathcal{A}} (\sum_{t}^{T} P_{F,\mu}(s_t = s)) \pi(s, a) f(s, a)
$$

$$
= T \cdot \sum_{s}^{\mathcal{S}} \sum_{a}^{\mathcal{A}} d_{F,\mu}(s) \pi(s, a) f(s, a)
$$

$$
= T \cdot \mathbb{E}_{d_{F,\mu}(s), \pi_F(s,a)}[f(s, a)] \qquad (37)
$$

where equation 36 holds in that $\forall s \notin \mathcal{S}_t : P(s_t = s) = 0$ and $\forall a \notin \mathcal{A}(s) : \pi(s, a) = 0$. By the fact that $S_t \cap S_{t'} = \emptyset$ for any $t \neq t'$ and any $\tau \in \mathcal{T}$ must pass some $s_t \in \mathcal{S}_t$ for all $t \in \{0, \ldots, T-1\}$, $P_{F,\mu}(s_t)$ is a valid distribution over $\mathcal{S}_t$ and $\sum_{s_t} P_{F,\mu}(s_t) = 1$. Accordingly, $d_{F,\mu}(s)$ is a valid distribution over $\mathcal{S}$ and $T \cdot d_{F,\mu}(s_t) = P_{F,\mu}(s_t)$. Analogically, we can define $d_{B,\rho} := \frac{1}{T} \sum_{t=1}^{T} P_{B,\rho}(s_t)$ such that for arbitrary function $f : \mathcal{S} \times \dot{\mathcal{A}} \rightarrow \mathbb{R}$,

$$
\mathbb{E}_{P_{B,\rho}(\tau)} \left[ \sum_{t=1}^{T} f(s_t, a_t) \right] = T \cdot \mathbb{E}_{d_{B,\rho}(s), \pi_F(s,a)}[f(s, a)] \qquad (38)
$$

### B.2 DAGs AS TRANSITION ENVIRONMENTS

**Theorem 3.** *(Golpar Raboky & Eftekhari, 2019): Let $P \in \mathbb{R}^{N \times N}$ be a non-negative matrix, the following statements are equivalent:*

   *1. $P$ is nilpolent;*

2. $P^N = 0$;

3. *The directed graph $\mathcal{G}(\mathcal{S}, \mathcal{A})$ associated with $P$ is a DAG graph;*

4. *There exists a permutation matrix $U$ such that $U^T P U$ is a strictly triangular matrix.*

where $\mathcal{S} = \{s^0, \dots, s^{N-1}\}$ and $\mathcal{A} = \{(s^i \to s^j) | P_{i,j} \neq 0\}$ are node and edge sets.

**Lemma 2.** *: For any DAG graph $\mathcal{G}(\mathcal{S}, \mathcal{A})$ associated with $P \in R^{N \times N}$ with $T + 1(\leq N)$ different topological node orders indexed by integers $[0, T]$,*

$$\forall t > T, \quad P^t = 0 \tag{39}$$

*Proof.* We prove the result by contradiction. Assuming $P^t(t > T)$ is not zero, then $\forall i \neq j$:

$$(P^t)_{i,j} = \sum_{k_{1:t-1}} P_{i,k_1} P_{k_1,k_2} \dots P_{k_{t-1},j} \tag{40}$$

By the nature of DAG graphs, $\forall (s' \to s) \in \mathcal{A} : s' \prec s$. Then the above expression is equal to:

$$
\begin{aligned}
&= \sum_{k_1 : s^i \prec s^{k_1}} P_{i,k_1} \left( \sum_{k_2 : s^{k_1} \prec s^{k_2}} P_{k_1,k_2} \dots \left( \sum_{k_{t-1} : s^{k_{t-2}} \prec s^{k_{t-1}}} P_{k_{t-2},k_{t-1}} P_{k_{t-1},j} \right) \right) \\
&= \sum_{k_{1:t-1} : (s^i \prec s^{k_1} \prec \dots \prec s^{k_{t-1}} \prec s^j)} P_{i,k_1} P_{k_1,k_2} \dots P_{k_{t-1},j} \\
&> 0
\end{aligned} \tag{41}
$$

Then it means that there at least exists a trajectory $(s^i \prec s^{k_1} \prec \dots \prec s^{k_{t-1}} < s^j)$ with non-zero probability. However, there are at least $t + 1$ distinct topological orders in the path, which contradicts the assumption that there are $T + 1$ different node orders. $\square$

Let's return to the graded DAG, $\mathcal{G}(\mathcal{S}, \mathcal{A})$ in GFlowNets. For the easiness of analysis, we restrict forward and backward policies and initial distribution to be tabular forms, $P_F \in \mathbb{R}^{|\mathcal{S}| \times |\mathcal{S}|}, \mu \in \mathbb{R}^{|\mathcal{S}|}$, $P_B \in \mathbb{R}^{|\mathcal{S}| \times |\mathcal{S}|}$, and $\rho \in \mathbb{R}^{|\mathcal{S}|}$ such that $P_F(s^j|s^i) = (P_F)_{j,i}$ and $P_B(s^j|s^i) = (P_B)_{j,i}$. Besides, we split initial distribution vectors by $\mu = [\bar{\mu}; 0] \in \mathbb{R}^{|\mathcal{S}|}$ and $\rho = [0; \bar{\rho}] \in \mathbb{R}^{|\mathcal{S}|}$, where $\bar{\mu}$ and $\bar{\rho}$ denote the probabilities of states except $s^f$ and $s^0$ respectively. We denote the graph equipped with self-loop over $s^f$ as $\mathcal{G}_F(\mathcal{S}, \mathcal{A} \cup \{(s^f \to s^f)\})$, and the reverse graph equipped with self-loop over $s^0$ as $\mathcal{G}_B(\mathcal{S}, \dot{\mathcal{A}} \cup \{(s^0 \to s^0)\})$. Accordingly, we enhanced $P_F$ and $P_B$ by defining $P_F(s^f|s^f) := 1$ and $P_B(s^0|s^0) := 1$. $(\mathcal{G}_F, P_F)$ specifies an absorbing Markov Chains: $s^f$ is the only absorbing state as only self-loop is allowed once entering $s^f$; the sub-graph over $S \setminus \{s^f\}$ is still a DAG, so any state $s \in S \setminus \{s^f\}$ is transient as it can be visited at least one time. Similarly, $(\mathcal{G}_B, P_B)$ specifies another absorbing Markov Chains with absorbing state $s^0$. For graph $\mathcal{G}_F$ and $\mathcal{G}_B$, their transition matrices $P_F$ and $P_B$ can be decomposed into:

$$P_F = \begin{pmatrix} \bar{P}_F & 0 \\ r_F & 1 \end{pmatrix}, P_B = \begin{pmatrix} 1 & r_B \\ 0 & \bar{P}_B \end{pmatrix} \tag{42}$$

Where $r_F^\top \in \mathbb{R}^{|\mathcal{S}|-1}$ and $r_B^\top \in \mathbb{R}^{|\mathcal{S}|-1}$ denotes the forward probability of $(s \to s^f)$ for any $s \in \mathcal{S} \setminus \{s^f\}$ and $(s^0 \leftarrow s)$, for any $s \in \mathcal{S} \setminus \{s^0\}$, $\bar{P}_F \in \mathbb{R}^{(|\mathcal{S}|-1) \times (|\mathcal{S}|-1)}$ and $\bar{P}_B \in \mathbb{R}^{(|\mathcal{S}|-1) \times (|\mathcal{S}|-1)}$ denote probability of $(s \to s')$ for any $s, s' \in \mathcal{S} \setminus \{s^f\}$ and $(s \leftarrow s')$ for any $s, s' \in \mathcal{S} \setminus \{s^0\}$.

**Lemma 3.** *: For $(\mathcal{G}, P_F, \mu)$ and $(\mathcal{G}_B, P_B, \rho)$, $d_{F,\mu} \in \mathbb{R}^{|\mathcal{S}|}$ and $d_{B,\rho} \in \mathbb{R}^{|\mathcal{S}|}$, can be written in the following form:*

$$d_{F,\mu} = (\bar{d}_{F,\mu}, 0), \quad \bar{d}_{F,\mu} = \frac{1}{T}(I - \bar{P}_F)^{-1}\bar{\mu} \tag{43}$$

$$d_{B,\rho} = (0, \bar{d}_{B,\rho}), \quad \bar{d}_{B,\rho} = \frac{1}{T}(I - \bar{P}_B)^{-1}\bar{\rho} \tag{44}$$

*Proof.* We prove the result for the forward case. A similar proof procedure can easily derived for the backward case, so it is omitted. First of all, by the nature of Markov Chains, $P_{F,\mu}(s_t = s^i) = [(P_F)^t \mu]_i$, and $d_{F,\mu} = \frac{1}{T} \sum_{t=0}^{T} (P_F)^t \mu$. Then, it can be easily verified (Grinstead & Snell, 2006):

$$(P_F)^t = \begin{pmatrix} (\bar{P}_F)^t & 0 \\ * & 1 \end{pmatrix}, \tag{45}$$

where the explicit expression of the upper right corner is omitted. By theorem 3, $\bar{P}_F$ is a nilpotent matrix and by lemma 2, $\frac{1}{T} \sum_{t=0}^{T} (\bar{P}_F)^t = \frac{1}{T} \sum_{t=0}^{\infty} (\bar{P}_F)^t = \frac{1}{T} (I - \bar{P}_F)^{-1}$(The second equality follows that fact that: $(I - \bar{P}_F) \sum_{t=0}^{\infty} (\bar{P}_F)^t = \sum_{t=0}^{\infty} (\bar{P}_F)^t - \sum_{t=1}^{\infty} (\bar{P}_F)^t = I$). Therefore,

$$d_{F,\mu} = \frac{1}{T} \begin{pmatrix} \sum_{t=0}^{T} \bar{P}_F^t & 0 \\ * & 1 \end{pmatrix} \mu = \left( \frac{1}{T} (I - \bar{P}_F)^{-1} \bar{\mu}, *\bar{\mu} \right)$$

By theorem 11.4 in Grinstead & Snell (2006), $(I - \bar{P}_F)_{i,j}^{-1}$ is the expected number of times the chain is in state $s_j$ starting from $s_i$ before absorbing in $s^f$. And $[(I - \bar{P}_F)^{-1} \bar{\mu}]_j$ is the expected number of times the chain is in state $s_j$ before absorbing. Since $\forall s \notin \mathcal{S}_0 : \mu(s) = 0$ and the fact $\mathcal{G}$ is graded, any trajectories over $\mathcal{G}$ must start from states in $\mathcal{S}_0$ to state in $\mathcal{S}_T$ i.e. $\sum_j [(I - \bar{P}_F)^{-1} \bar{\mu}]_j = T$. Thus, $\frac{1}{T} [(I - \bar{P}_F)^{-1} \bar{\mu}]_i$ denote the fraction of staying in transient state $s_i$ before absorbing, this is, the probability observing state $s_i$ within $T$ time steps. By the same reasoning, we can conclude that $*\bar{\mu} = 0$ as $s^f$ can not be reached within $T - 1$ transitions. □

**Lemma 4.** *: For two forward policy, $\pi_F$ and $\pi_F'$, and two backward policy, $\pi_B$ and $\pi_{B'}$, we have:*

$$\begin{aligned} D_{TV}(d_{F,\mu}(\cdot), d_{F,\mu}'(\cdot)) &\leq D_{TV}^{d_{F,\mu}'}(\pi_F(s,\cdot), \pi_F'(s,\cdot)), \\ D_{TV}(d_{B,\rho}(\cdot), d_{B,\rho}'(\cdot)) &\leq D_{TV}^{d_{B,\rho}'}(\pi_B(s,\cdot), \pi_B'(s,\cdot)), \end{aligned} \tag{46}$$

*where for three arbitrary distributions p,q and u, $D_{TV}(p(\cdot), q(\cdot)) := \frac{1}{2} ||p(\cdot) - q(\cdot)||_1$ and $D_{TV}^u(p(\cdot|s), q(\cdot|s)) := \frac{1}{2} \mathbb{E}_{u(s)} ||p(\cdot|s) - q(\cdot|s)||_1$.*

*Proof.* The proof procedure follows that of Lemma 3 in Achiam et al. (2017). For two forward policy $\pi_F$ and $\pi_F'$, denoting $\bar{N}_F := (I - \bar{P}_F)^{-1}$ and $\bar{N}_F' := (I - \bar{P}_F')^{-1}$, then

$$\bar{N}_F^{-1} - (\bar{N}_F')^{-1} = \bar{P}_F' - \bar{P}_F := \Delta \tag{47}$$

and

$$\bar{N}_F' - \bar{N}_F = \bar{N}_F' \Delta \bar{N}_F \tag{48}$$

Then,

$$\begin{aligned} ||d_{F,\mu} - d_{F,\mu}'||_1 &= ||\bar{d}_{F,\mu} - \bar{d}_{F,\mu}'||_1 \\ &= \frac{1}{T} ||(\bar{N}_F - \bar{N}_F') \bar{\mu}||_1 = \frac{1}{T} ||\bar{N}_F \Delta \bar{d}_{F,\mu}'||_1 \\ &\leq \frac{1}{T} ||\bar{N}_F||_1 ||\Delta \bar{d}_{F,\mu}'||_1 \leq ||\Delta \bar{d}_{F,\mu}'||_1 \end{aligned} \tag{49}$$

Therefore, we have

$$\begin{aligned} ||\Delta \bar{d}_{F,\mu}'||_1 &= ||(P_F' - P_F) d_{F,\mu}'||_1 = \sum_s \left| \sum_{s'} (P_F'(s'|s) - P_F(s'|s)) d_{F,\mu}'(s) \right| \\ &\leq \sum_{s,s'} |P_F'(s'|s) - P_F(s'|s)| d_{F,\mu}'(s) = \sum_{s,a} |\pi'(s,a) - \pi(s,a)| d_{F,\mu}'(s) \\ &= \mathbb{E}_{d_{F,\mu}'(s)} [||\pi_F'(s,\cdot) - \pi_F(s,\cdot)||_1] \end{aligned}$$

Where first equality holds by the fact that $P_F(s^f|s^f) = P_F'(s^f|s^f) = 1$ for any two forward policies consistent with $\mathcal{G}_F$. The result of backward policies can be derived analogically and is omitted here. □

### B.3 DERIVATION OF GRADIENTS

**Proposition 4.** *The gradients of $J_F(\theta)$ and $J_B^\phi$ w.r.t $\theta$ and $\phi$ can be written as:*

$$
\begin{aligned}
\nabla_\theta J_F(\theta) &= T\mathbb{E}_{d_{F,\mu}(s)\pi_F(s,a)}\left[Q_F(s,a)\nabla_\theta log\pi_F(s,a;\theta)\right] + \mathbb{E}_{\mu(s_0)}[V_F(s_0)\nabla_\theta log\mu(s_0;\theta)] \\
&= T\mathbb{E}_{d_{F,\mu}(s)\pi_F(s,a)}\left[A_F(s,a)\nabla_\theta log\pi_F(s,a;\theta)\right] + \mathbb{E}_{\mu(s_0)}[V_F(s_0)\nabla_\theta log\mu(s_0;\theta)] \\
\nabla_\phi J_B(\phi) &= T\mathbb{E}_{d_{B,\rho}(s)\pi_B(s,a)}\left[Q_B(s,a)\nabla_\phi log\pi_B(s,a;\phi)\right] \\
&= T\mathbb{E}_{d_{B,\rho}(s)\pi_B(s,a)}\left[A_B(s,a)\nabla_\phi log\pi_B(s,a;\phi)\right]
\end{aligned}
\tag{50}
$$

*Proof.*

$$
\nabla_\theta J_F(\theta) = \nabla_\theta \frac{Z^\theta}{\hat{Z}^\theta}V_F(s_0;\theta) = \frac{V_F(s_0;\theta)}{\hat{Z}^\theta} + \underbrace{\frac{Z^\theta}{\hat{Z}^\theta}\nabla_\theta V_F(s_0;\theta)}_{(0)}
$$

$$
\begin{aligned}
&\overset{(0)}{=} \mathbb{E}_{P_{F,\mu}(s_0)}[\nabla_\theta V_F(s_0;\theta)] = \mathbb{E}_{P_{F,\mu}(s_0)}\left[\nabla_\theta \mathbb{E}_{\pi_F(s_0,a_0;\theta)}[Q_F(s_0,a_0;\theta)]\right] \\
&= \mathbb{E}_{P_{F,\mu}(s_0,a_0)}\left[\nabla_\theta log\pi_F(s_0,a_0;\theta)Q_F(s_0,a_0;\theta) + \nabla_\theta Q_F(s_0,a_0;\theta)\right] \\
&= \mathbb{E}_{P_{F,\mu}(s_0,a_0)}\left[\nabla_\theta log\pi_F(s_0,a_0;\theta)Q_F(s_0,a_0;\theta)\right] + \underbrace{\mathbb{E}_{P_{F,\mu}(s_0,a_0)}\left[\nabla_\theta R_F(s_0,a_0;\theta) + V_F(s_1;\theta)\right]}_{(1)}
\end{aligned}
$$

$$
\overset{(1)}{=} \underbrace{\mathbb{E}_{P_{F,\mu}(s_0,a_0)}\left[\nabla_\theta log\frac{\pi_F(s_0,a_0;\theta)}{\pi_B(s_1,a_0)}\right]}_{(2)} + \mathbb{E}_{P_{F,\mu}(s_1)}\left[\nabla_\theta V_F(s_1;\theta)\right]
$$

$$
\begin{aligned}
&\overset{(2)}{=} \mathbb{E}_{P_{F,\mu}(s_0)}\left[\mathbb{E}_{\pi_F(s_0,a_0)}[\nabla_\theta log\pi_F(s_0,a_0;\theta)]\right] \\
&= \underbrace{\mathbb{E}_{P_{F,\mu}(s_0)}[\nabla_\theta 1] = 0}_{\text{By Lemma1}}
\end{aligned}
\tag{51}
$$

Therefore,

$$
\mathbb{E}_{P_{F,\mu}(s_0)}[\nabla_\theta V_F(s_0;\theta)] = \mathbb{E}_{P_{F,\mu}(s_0,a_0)}\left[\nabla_\theta log\pi_F(s_0,a_0;\theta)Q_F(s_0,a_0;\theta)\right] + \mathbb{E}_{P_{F,\mu}(s_1)}\left[\nabla_\theta V_F(s_1;\theta)\right]
\tag{52}
$$

Keep doing the process, we have

$$
\mathbb{E}_{P_{F,\mu}(s_t)}[\nabla_\theta V_F(s_t;\theta)] = \mathbb{E}_{P_{F,\mu}(s_t,a_t)}\left[\nabla_\theta log\pi_F(s_t,a_t;\theta)Q_F(s_t,a_t;\theta)\right] + \mathbb{E}_{P_{F,\mu}(s_{t+1})}\left[\nabla_\theta \underbrace{V_F(s_{t+1};\theta)}_{V(s_T)=0}\right]
\tag{53}
$$

Then,

$$
\begin{aligned}
\nabla_\theta J_F(\theta) &= \mathbb{E}_{P_{F,\mu}(\tau)}\left[\sum_{t=0}^{T-1}\nabla_\theta log\pi_F(s_t,a_t;\theta)Q_F(s_t,a_t;\theta)\right] \\
&= \mathbb{E}_{d_{F,\mu}(s)\pi_F(s,a)}\left[\nabla_\theta log\pi_F(s,a;\theta)Q_F(s,a;\theta)\right]
\end{aligned}
\tag{54}
$$

Besides,

$$
\begin{aligned}
\nabla_\theta J_F(\theta) &= \mathbb{E}_{d_{F,\mu}(s)\pi_F(s,a)}\left[\nabla_\theta log\pi_F(s,a;\theta)Q_F(s,a;\theta)\right] - \mathbb{E}_{d_{F,\mu}(s)}\left[V_F(s)\underbrace{\mathbb{E}_{\pi_F(s,a)}[\nabla_\theta log\pi_F(s,a;\theta)]}_{=0}\right] \\
&= \mathbb{E}_{d_{F,\mu}(s)\pi_F(s,a)}\left[\nabla_\theta log\pi_F(s,a;\theta)A_F(s,a;\theta)\right]
\end{aligned}
\tag{55}
$$

$\square$

## B.4 CONNECTION OF POLICY-BASED TRAINING TO TB-BASED TRAINING

As shown in Appendix B.3, the gradient of $J_F$ w.r.t $\theta_F$ can be written as:

$$
\nabla_{\theta_F} J_F(\theta_F) = T \mathbb{E}_{d_{F,\mu}(s)\pi_F(s,a)} \left[ Q_F(s,a) \nabla_{\theta_F} log \pi_F(s,a;\theta_F) \right]
$$

$$
= \mathbb{E}_{P_{F,\mu}(\tau)} \left[ \sum_{t=0}^{T-1} \nabla_\theta log \pi_F(s_t,a_t;\theta) Q_F(s_t,a_t) \right]
$$

$$
= \mathbb{E}_{P_{F,\mu}(\tau)} \left[ \sum_{t=0}^{T-1} \nabla_\theta log \pi_F(s_t,a_t;\theta) \mathbb{E}_{P_{F,\mu}(\tau)} \left[ \sum_{t'=t}^{T-1} R_F(s_{t'},a_{t'}) \middle| s_t,a_t \right] \right]
$$

$$
= \mathbb{E}_{P_{F,\mu}(\tau)} \left[ \sum_{t=0}^{T-1} \nabla_\theta log \pi_F(s_t,a_t;\theta) \left( \sum_{t'=t}^{T-1} R_F(s_{t'},a_{t'}) \right) \right]
$$

$$
- C \underbrace{\mathbb{E}_{P_{F,\mu}(\tau)} \left[ \sum_{t=0}^{T-1} \nabla_\theta log \pi_F(s_t,a_t;\theta) \right]}_{=0 \text{ by Lemma 1}}
$$

$$
= \mathbb{E}_{P_{F,\mu}(\tau)} \left[ \sum_{t=0}^{T-1} \nabla_\theta log \pi_F(s_t,a_t;\theta) \left( \sum_{t'=t}^{T-1} R_F(s_{t'},a_{t'}) - C \right) \right] \tag{56}
$$

where $C$ is some added baseline and constant w.r.t $\theta$ for variance reduction during gradient estimation. Given a batch of $\mathcal{D} = \{\tau^1, \ldots, \tau^N\}$, this result implies that: when we estimate $\nabla_{\theta_F} J_F(\theta_F)$ directly without the help of $Q_F$ and $V_F$ (it is equivalent to estimate $\nabla_{\theta_F} \mathcal{L}_{TB}(\theta_F)$ or $\nabla_{\theta_F} D_{KL}^\mu(P_F(\tau|s_0;\theta_F), \tilde{P}_B(\tau|s_0))$ by Proposition 1), we approximate $Q_F(s,a)$ empirically by $\frac{1}{N_{s,a}} \sum_{n:(s_t^n,a_t^n)=(s,a)} \sum_{t'=t}^{T-1} R_F(s_{t'}^n, a_{t'}^n;\theta)$, where $N_{s,a}$ is the number of trajectories that pass $(s,s')$ at time $t$ ($a := (s \rightarrow s')$), and reduce the estimation variance by a constant $C$ that is computed based on current data batch $\mathcal{D}$. By comparison, under the RL formulation, $\nabla_{\theta_F} J_F(\theta_F)$ is computed by $\mathbb{E}_{P_{F,\mu}(\tau)} \left[ \sum_{t=0}^{T-1} \nabla_\theta log \pi_F(s_t,a_t;\theta) (Q_F(s_t,a_t) - V_F(s_t)) \right]$, where $V_F(s')$ is approximated by a parameterized function $\tilde{V}_F(s',\eta)$ (see Appendix B.5), and $Q(s,a)$ is approximated by $R(s,a) + \tilde{V}_F(s')$. This functional approximation of $Q(s,a)$ leads to biased gradient estimation but typically reduces variance Schulman et al. (2016). The constant baseline $C$ is generalized to an unbiased functional baseline $\tilde{V}_F$, which utilizes not only the current data $\mathcal{D}$ but also all the past data.

Based on the discussion above, $Q_F$ and $V_F$ play the role of reducing the variance of gradient estimation. In principle, both TB-based training and policy-based training correspond to minimizing $D_{KL}^\mu(P_F(\tau|s_0;\theta_F), \tilde{P}_B(\tau|s_0))$, this support the stability our policy-based method with non-stationary reward.

## B.5 MODEL PARAMETER UPDATING RULES

We list the model updating rules for $P_F$ and $\mu$ below. The updating rules for $P_B$ follow analogically. Denoting the updated parameter and current parameter as $\theta'$ and $\theta$ respectively, the first way for model updating is by the following objective

$$
\min_\theta T \cdot \mathbb{E}_{d_{F,\mu}(s),\pi_F(s,a;\theta)} [A_F(s,a)] + \mathbb{E}_{\mu(s_0;\theta)}[V_F(s_0)] \tag{57}
$$

The gradients of the above formula are equal to $\nabla_\theta J_F(\theta)$. Given a batch of $\mathcal{D} = \{\tau^1, \ldots, \tau^N\}$, the gradient is approximated by:

$$
\hat{g}_F := \frac{1}{N} \sum_{n=1}^{N} \sum_{t=0}^{T-1} \left[ (\hat{Q}_F(s_t^n,a_t^n) - \tilde{V}_F(s_t;\eta)) \nabla_\theta log \pi_F(s_t^n,a_t^n;\theta) \right] + \frac{1}{N} \sum_{n=1}^{N} \hat{V}_F(s_0^n) \frac{\nabla_\theta Z_\theta}{\hat{Z}} \tag{58}
$$

In the equation above, $\hat{V}_F(s_t^n) = \sum_{t'=t}^{T-1} R(s_{t'}^n, a_{t'}^n)$; $\tilde{V}_F(\cdot;\eta)$ is the functional approximation to $V_F(\cdot)$ by objective $\mathbb{E}_{d_{F,\mu}(s)}[(V_F(s) - \tilde{V}_F(s;\eta))^2]$ whose gradient w.r.t $\eta$ are approximated by $\hat{g}_V := \nabla_\eta \frac{1}{N} \sum_{n,t} (\hat{V}_F(s_t^n) - \tilde{V}_F(s_t^n))^2$; $\hat{Q}_F(s_t^n,a_t^n) = \sum_{t'=t}^{T-1} R(s_{t'}^n, a_{t'}^n)$, which is a unbiased estimation

for $Q_F(s_t^n, a_t^n)$ ( or $= R_F(s_t^n, a_t^n) + \tilde{V}_F(s_t^n; \eta)$), which is biased estimation but typically has lower variance.). Finally, the model parameters are updated by $\theta' \leftarrow \theta - \alpha_F \hat{g}_F$ and $\eta' \leftarrow \eta - \alpha_V \hat{g}_V$.

For the trust-region way, the optimization objective is defined as:

$$
\begin{aligned}
\min_{\theta'_F, \theta'_Z} \quad & T \cdot \mathbb{E}_{d_{F,\mu}(s;\theta), \pi(s,a;\theta'_F)}[A_F(s,a;\theta)] + \mathbb{E}_{\mu(s_0;\theta'_Z)}[V_F(s_0;\theta)] \\
\text{s.t.} \quad & D_{\mathrm{KL}}^{d_{F,\mu}(\cdot;\theta)}(\pi_F(s,a;\theta_F), \pi_F(s,a;\theta'_F)) \leq \delta_F
\end{aligned}
\tag{59}
$$

In practice, the above formula can be linearly approximated by:

$$
\begin{aligned}
\min_{\theta'_F, \theta'_Z} \quad & \left(T \cdot \nabla_{\theta'_F} \mathbb{E}_{d_{F,\mu}(s;\theta), \pi(s,a;\theta'_F)}[A_F(s,a;\theta)]\right)^\top (\theta'_F - \theta_F) + \mathbb{E}_{\mu(s_0;\theta'_Z)}[V_F(s_0;\theta)] \\
\text{s.t.} \quad & (\theta'_F - \theta_F)^\top \left(\nabla^2_{\theta'_F} D_{\mathrm{KL}}^{d_{F,\mu}(\cdot;\theta)}(\pi_F(s,a;\theta_F), \pi_F(s,a;\theta'_F))\right)(\theta'_F - \theta_F) \leq \delta_F
\end{aligned}
\tag{60}
$$

Denoting $\hat{g}_F$ and $\hat{H}$ as the estimation of the first-order gradient of the objective and the second-order gradient of the KL divergence constraint w.r.t $\theta_F$, $\hat{g}_Z$ as the estimation of the first-order gradient of the objective w.r.t $\theta_Z$, then model parameters are updated by:

$$
\theta'_F \leftarrow \theta_F - \left(\frac{2\delta_F}{\hat{g}^\top \hat{H}^{-1} \hat{g}}\right)^{0.5} \hat{H}^{-1} \hat{g}, \quad \theta'_Z \leftarrow \theta_Z - \alpha_Z \hat{g}_Z
\tag{61}
$$

When the dimension of $\theta'$ is high, compute $\hat{H}^{-1}$ is computationally expensive, so we use the conjugate gradient method which can estimate $\hat{H}^{-1} \hat{g}_F$ based on $\hat{H} \hat{g}_F$ (Hestenes et al., 1952).

## C  PERFORMANCE ANALYSIS

### C.1  PROOF OF THEOREM 1

*Proof.*

$$
\begin{aligned}
J_F^G &= J_F + (J_F^G - J_F) \\
&= J_F + \mathbb{E}_{P_{F,\mu}(\tau)}\left[log\frac{P_F(\tau|s_0)Z}{P_G(\tau|x)R(x)} - log\frac{P_F(\tau|s_0)Z}{P_B(\tau|x)R(x)}\right] \\
&= J_F + \mathbb{E}_{P_{F,\mu}(\tau)}\left[log\frac{P_B(\tau|x)}{P_G(\tau|x)}\right] + \mathbb{E}_{P_{B,\rho}(\tau)}\left[log\frac{P_B(\tau|x)}{\tilde{P}_G(\tau|x)}\right] - \mathbb{E}_{P_{B,\rho}(\tau)}\left[log\frac{P_B(\tau|x)}{\tilde{P}_G(\tau|x)}\right] \\
&= J_F + J_B^G + \mathbb{E}_{\rho(x)}[logZ_G(x)] + \sum_\tau (P_{F,\mu}(\tau) - P_{B,\rho}(\tau))R_B^G(\tau|x)
\end{aligned}
\tag{62}
$$

where $R_B^G(\tau|x) := log\frac{P_B(\tau|x)}{P_G(\tau|x)} = \sum_{t=1}^T R_B^G(s_t, a_t)$, then

$$
\begin{aligned}
J_F^G &= J_F + J_B^G + \langle P_{F,\mu}(\cdot) - P_{B,\rho}(\cdot), R_B^G(\cdot)\rangle \\
&\leq J_F + J_B^G + ||P_{F,\mu}(\cdot) - P_{B,\rho}(\cdot)||_1 ||R_B^G(\cdot)||_\infty \\
&\leq J_F + J_B^G + T \cdot ||P_{F,\mu}(\cdot) - P_{B,\rho}(\cdot)||_1 R_B^{G,max}
\end{aligned}
\tag{63}
$$

where the first inequality follows from Hölder's inequality and the second inequality holds by $max_\tau R_B^G(\tau) \leq T \cdot \max_{s,a} R_B^G(s,a) := T \cdot R_B^{G,max}$. By Pinsker's inequality:

$$
||P_{F,\mu}(\cdot) - P_{B,\rho}(\cdot)||_1 \leq \sqrt{\frac{1}{2}D_{\mathrm{KL}}(P_{F,\mu}(\tau), P_{B,\rho}(\tau))}
\tag{64}
$$

Besides,

$$
\begin{aligned}
D_{\mathrm{KL}}(P_{F,\mu}(\tau), P_{B,\rho}(\tau)) &= \mathbb{E}_{P_{F,\mu}(\tau)}\left[log\frac{P_F^\mu(\tau|x)P_{F,\mu}^\top(x)}{P_B(\tau|x)P_{F,\mu}^\top(x)}\right] \\
&\leq \mathbb{E}_{P_{F,\mu}(\tau)}\left[log\frac{P_F^\mu(\tau|x)}{P_B(\tau|x)}\right] + \underbrace{\mathbb{E}_{P_{F,\mu}^\top(x)}\left[log\frac{P_{F,\mu}^\top(x)}{R(x)/Z^*}\right]}_{\geq 0} \\
&= D_{\mathrm{KL}}(P_{F,\mu}(\tau), P_B(\tau)) = D_{\mathrm{KL}}^\mu(P_F(\tau), \tilde{P}_B(\tau)) - logZ + logZ^* \\
&= J_F + logZ^* - logZ
\end{aligned}
\tag{65}
$$

Then we have:

$$
J_F^G \leq J_F + J_B^G + \mathbb{E}_{\rho(x)}[logZ_G(x)] + R_B^{G,max}\sqrt{\frac{1}{2}(J_F + logZ - logZ^*)}
\tag{66}
$$

$\square$

**Lemma 5.** *Given two forward policies $(\pi_F, \pi_F')$ or backward $(\pi_B, \pi_B')$, we have*

$$
\begin{aligned}
\frac{1}{T}(J_F - J_F') &= \mathbb{E}_{d_{F,\mu}(s),\pi(a,s)}[A_F'(s,a)] + D_{\mathrm{KL}}^{d_{F,\mu}}(\pi_F(s,a), \pi_F'(s,a)) \\
\frac{1}{T}(J_B - J_B') &= \mathbb{E}_{d_{B,\rho}(s),\pi(a,s)}[A_B'(s,a)] + D_{\mathrm{KL}}^{d_{F,\rho}}(\pi_B(s,a), \pi_B'(s,a))
\end{aligned}
\tag{67}
$$

*Proof.*

$$
\begin{aligned}
V_F(s_0) - V_F'(s_0) &= \mathbb{E}_{P_F(\tau|s_0)}\left[\sum_{t=0}^{T-1}R_F(s_t,a_t)\right] + \underbrace{V_F'(s_T)}_{:=0} - V_F'(s_0) \\
&= \mathbb{E}_{P_F(\tau|s_0)}\left[\sum_{t=0}^{T-1}R_F(s_t,a_t) + V_F'(s_t) - V_F'(s_t)\right] - V_F'(s_0) + V_F'(s_T) \\
&= \mathbb{E}_{P_F(\tau|s_0)}\left[\sum_{t=0}^{T-1}R_\pi(s_t,a_t) + V_F'(s_{t+1}) - V_F'(s_t)\right] \\
&= \mathbb{E}_{P_F(\tau|s_0)}\left[\sum_{t=0}^{T-1}A_F'(s_t,a_t)\right] + \mathbb{E}_{P_F(\tau|s_0)}\left[\sum_{t=0}^{T-1}(R_F(s_t,a_t) - R_F'(s_t,a_t))\right]
\end{aligned}
\tag{68}
$$

Thus,

$$
\begin{aligned}
J_F - J_F' &= \mathbb{E}_{\mu(s_0)}[V_F(s) - V_F'(s)] \\
&= \mathbb{E}_{P_{F,\mu}(\tau)}\left[\sum_{t=0}^{T-1}A_F'(s_t,a_t)\right] + \mathbb{E}_{P_{F,\mu}(\tau)}\left[\sum_{t=0}^{T-1}(R_F(s_t,a_t) - R_F'(s_t,a_t))\right] \\
&= \mathbb{E}_{d_{F,\mu}(s),\pi_F(s,a)}[A_F'(s,a)] + \mathbb{E}_{d_{F,\mu}(s)\pi(a,s)}[R_F(s,a) - R_F'(s,a)] \\
&= T \cdot \mathbb{E}_{d_{F,\mu}(s),\pi_F(s,a)}[A_F'(s,a)] + T \cdot \mathbb{E}_{d_{F,\mu}(s)}[D_{\mathrm{KL}}(\pi_F(\cdot,s), \pi_F'(\cdot,s))]
\end{aligned}
\tag{69}
$$

The result for the backward case can easily be derived following a similar proof procedure as the forward case. $\square$

### C.2 PROOF OF THEOREM 2

By Lemma 5:

$$
\frac{1}{T}(J_F - J_F') = \mathbb{E}_{d_{F,\mu}(s),\pi(a,s)}[A_F'(s,a)] + \underbrace{D_{\mathrm{KL}}^{d_{F,\mu}}(\pi_F(\cdot,s), \pi_F'(\cdot,s))}_{=\delta_F}
\tag{70}
$$

Let $\bar{A}'_F \in R^{|\mathcal{S}|}$ denote the vector components of $\mathbb{E}_{\pi_F(s,a)}[A'_F(s,a)]$. Then we have

$$
\begin{aligned}
\mathbb{E}_{d_{F,\mu}(s)\pi_F(s,a)}[A'_F(s,a)] &= \langle d_{F,\mu}, \bar{A}'_F \rangle \\
&= \langle d'_{F,\mu}, \bar{A}'_F \rangle + \langle d_{F,\mu} - d'_{F,\mu}, \bar{A}'_F \rangle \\
&\leq \mathbb{E}_{d'_{F,\mu}(s)\pi_F(s,a)}[A'_F(s,a)] + ||d_{F,\mu} - d'_{F,\mu}||_1 ||\bar{A}'_F||_\infty
\end{aligned}
\tag{71}
$$

By Lemma 4:

$$
\leq \mathbb{E}_{d'_{F,\mu}(s)\pi_F(a,s)}[A'_F(s,a)] + 2\mathbb{E}_{d'_{F,\mu}(s)}\big[D_{TV}(\pi_F(s,\cdot), \pi'_F(s,\cdot))\big]\epsilon'_F
\tag{72}
$$

By Pinsker's inequality:

$$
\leq \mathbb{E}_{d'_{F,\mu}(s)\pi_F(a,s)}[A'_F(s,a)] + 2\mathbb{E}_{d'_{F,\mu}(s)}\left[\left(\frac{1}{2}D_{KL}(\pi(\cdot,s), \pi'(\cdot,s))\right)^{0.5}\right]\epsilon'_F
\tag{73}
$$

By Jensen's inequality:

$$
\leq E_{d'_{F,\mu}(s)\pi(a,s)}[A'_F(s,a)] + 2\left(\frac{1}{2}\mathbb{E}_{d'_{F,\mu}(s)}[D_{KL}(\pi(\cdot,s), \pi'(\cdot,s))]\right)^{0.5}\epsilon'_F
\tag{74}
$$

Thus, we have

$$
\frac{1}{T}(J_F - J'_F) \leq E_{d'_{F,\mu}(s)\pi(a,s)}[A'_F(s,a)] + (2\delta)^{0.5}\epsilon'_F + \delta_F
\tag{75}
$$

## D  ADDITIONAL EXPERIMENT INFORMATION

In all three domains, we follow a regular way of sample policy design: sample policy is a mix of the learned forward policy and a uniform policy where the mix-in factor of the uniform policy starts at $0.5$ and decays exponentially at a rate of $0.99$ after each training iteration. We use the Adam optimizer with learning rate $\{10^{-3}, 5 \times 10^{-3}, 10^{-1}\}$ for the optimization of policies, value functions, and total flow estimator respectively. The batch size is set to $128$.

### D.1  HYPER-GRID MODELING

For value-based methods like TB-based training strategy, the policy $P_\mathcal{D}$ for training data sampling works in an offline way. The design of guided distribution follows the way proposed by Shen et al. (2023). Explicit designs are shown in the following sections.

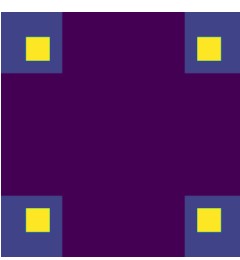

**Environment** In this hyper-grid environment, $\mathcal{S} \setminus \{s^f\}$ is equal to $\{s = ([s]_1, \ldots, [s]_D) | \forall i \in [1, \ldots, D], [s]_i \in [0, \ldots, N-1]\}$ where the initial state $s^0 = (0, \ldots, 0)$ and $s^f$ can be represented by any invalid coordinate tuple of the hyper-grid, and is denoted as $(-1, \ldots, -1)$ in our implementation. For each state $s \neq s^f$, we have $D + 1$ possible actions in $\mathcal{A}(s)$: (1) increment the $i_{th}$ coordinate by one, arriving at $s' = ([s]_0, \ldots, [s]_i + 1, \ldots)$; (2) choose stopping actions $(s \rightarrow s^f)$, terminating the process and return $x = s$ as the terminating coordinate tuple. By definition, $\mathcal{G}$ is not a graded DAG, and $\mathcal{S} \setminus \{s^f\} = \mathcal{X}$ as all coordinate tuples can be returned as the terminating states. The reward $R(x)$ is defined as:

Figure 6: Graphical representation of the reward function for a two-dimensional hyper-grid.

$$
R(x) = R_0 + R_1 \prod_{d=1}^{D} \mathbf{1}\left[\left|\frac{[s]_d}{N-1} - 0.5\right| \in (0.25, 0.5]\right] + R_2 \prod_{d=1}^{D} \mathbf{1}\left[\left|\frac{[s]_d}{N-1} - 0.5\right| \in (0.3, 0.4]\right]
\tag{76}
$$

|  | 256x256 | | | 128x128x128 | | | 32x32x32x32 | | |
|---|---|---|---|---|---|---|---|---|---|
| Method | $D_{TV}$ | $D_{JSD}$ | $T_{train}$ | $D_{TV}$ | $D_{JSD}$ | $T_{train}$ | $D_{TV}$ | $D_{JSD}$ | $T_{train}$ |
| DB-U | $0.573 \pm 0.016$ | $0.220 \pm 0.012$ | 40.5 | $0.429 \pm 0.014$ | $0.185 \pm 0.011$ | 36.3 | $0.376 \pm 0.030$ | $0.088 \pm 0.003$ | 11.4s |
| TB-U | $0.208 \pm 0.075$ | $0.071 \pm 0.039$ | 24.5 | $0.498 \pm 0.001$ | $0.215 \pm 0.001$ | 23.1 | $0.098 \pm 0.007$ | $0.012 \pm 0.002$ | 20.4 |
| RL-U | $0.104 \pm 0.053$ | $0.031 \pm 0.013$ | 33.4 | $0.323 \pm 0.214$ | $0.131 \pm 0.102$ | 23.8 | $0.074 \pm 0.010$ | $0.008 \pm 0.001$ | 21.0 |
| RL-T | $0.194 \pm 0.078$ | $0.060 \pm 0.042$ | 71.6 | $\mathbf{0.052} \pm 0.003$ | $\mathbf{0.005} \pm 0.000(2)$ | 53.4 | $0.101 \pm 0.003$ | $0.019 \pm 0.002$ | 59.3 |
| DB-B | $0.403 \pm 0.018$ | $0.128 \pm 0.022$ | 53.0 | $0.247 \pm 0.013$ | $0.056 \pm 0.006$ | 46.2 | $0.101 \pm 0.012$ | $0.020 \pm 0.003$ | 14.1 |
| TB-B | $0.351 \pm 0.142$ | $0.129 \pm 0.081$ | 29.6 | $0.066 \pm 0.036$ | $0.008 \pm 0.005$ | 21.7 | $0.101 \pm 0.013$ | $0.012 \pm 0.001$ | 20.6 |
| RL-B | $0.100 \pm 0.093$ | $0.028 \pm 0.041$ | 43.3 | $0.057 \pm 0.022$ | $0.006 \pm 0.002$ | 43.2 | $0.079 \pm 0.009$ | $0.007 \pm 0.001$ | 25.2 |
| RL-G | $\mathbf{0.086} \pm 0.073$ | $\mathbf{0.016} \pm 0.025$ | 43.1 | $0.055 \pm 0.021$ | $0.006 \pm 0.002$ | 42.8 | $\mathbf{0.056} \pm 0.007$ | $\mathbf{0.005} \pm 0.001$ | 24.4 |

Table 1: Hyper-grid modeling performance comparison of GFlowNets trained by different strategies.

where $R_0 = 10^{-2}$, $R_1 = 0.5$ and $R_2 = 2$ in our experiment. For a trajectory $\tau = (s_0, \ldots, s_i, \ldots, s_f)$, the probability of the guided distribution can be written as:

$$P_G(\tau|x) = \prod_i P_G(s_i|s_{i-1}, x),$$

$$\forall s_i \neq s^f, \quad P_G(s_i|s_{i-1}, x) = \begin{cases} \frac{P_F(s_i|s_{i-1})}{\sum_{s:s \neq s^f} P_F(s|s_{i-1}) + \epsilon^f}, & \text{if } R(s_{i-1}) \leq R_0 \\ P_F(s_i|s_{i-1}) & \text{otherwise} \end{cases}$$

$$P_G(s^f|s_{i-1}, x) = \begin{cases} \frac{\epsilon^f}{\sum_{s:s \neq s^f} P_F(s|s_{i-1}) + \epsilon^f}, & \text{if } R(s_{i-1}) \leq R_0 \\ P_F(s^f|s_{i-1}) & \text{otherwise} \end{cases} \quad (77)$$

where $s_i \neq s^f$ and $\epsilon^f = 10^{-5}$. As all the coordinate tuples can be terminating states, this is, $s_f$ is the child of all the other states, the expression above means that for a state $s_{i-1}$ with low reward, its probability of being a terminating state is replaced by a small value $\epsilon^f$. In this way, we discourage the generative process from stopping early at low reward coordinate tuples.

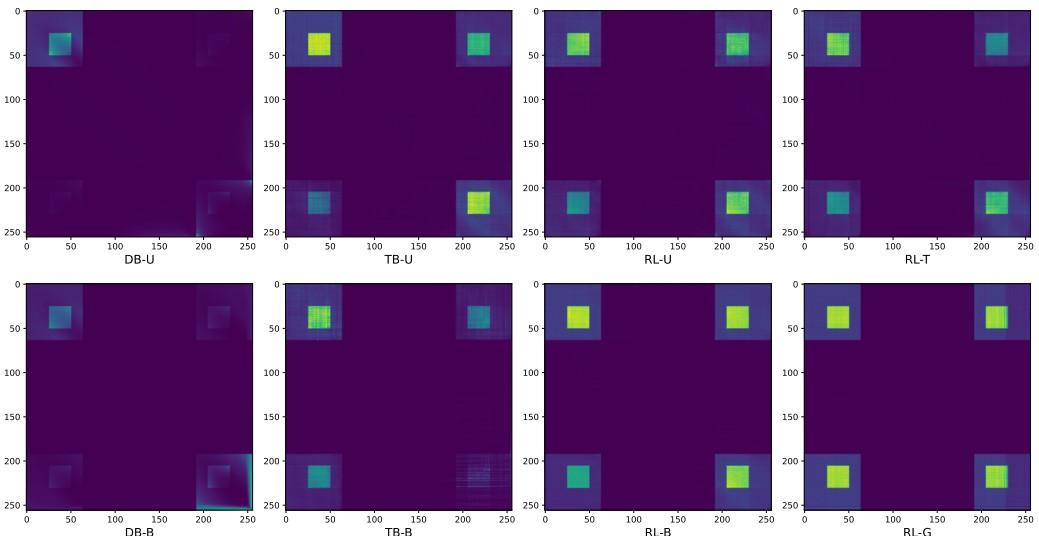

Figure 7: Graphical illustrations of $P_F^\top(x)$ averaged across 5 runs of different training strategies for a $256 \times 256$ hyper-grid.

**Model Architecture** Policy $P_F$ is parametrized by a neural network with 4 hidden layers and the hidden dimension is 256. Policy $P_B$ is fixed to be uniform over valid actions or parameterized in the same way as $P_F$. Coordinate tuples are transformed by K-hot encoding before being fed into Neural Networks.

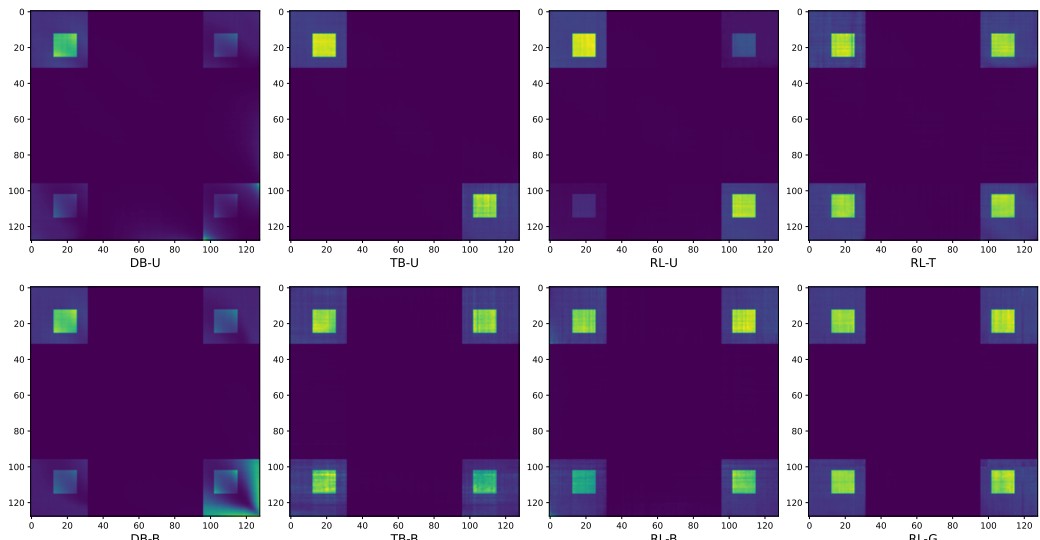

Figure 8: Graphical illustrations of $P_F^\top(x)$ averaged across 5 runs of different training strategies for a $128 \times 128 \times 128$ hyper-grid. For visualization easiness, only the marginals of 2 dimensions are plotted.

## D.2 SEQUENCE DESIGN

**Environment** In this environment, $\mathcal{S} = \{-1, 0, \ldots, N-1\}^D$ with element $s$ corresponds to a sequence composed of integers ranging from $-1$ and $N-1$ logits. The set $\{0, \ldots, N-1\}$ denotes the nucleotide types or building blocks, and the integer $-1$ represents that the corresponding position within $s$ is unfilled.

The initial empty sequence $s^0$ is represented by $\{-1\}^D$. For $s_t \in \mathcal{S}_t$, there are $t$ elements in $\{0, \ldots, N-1\}$ and the rest equal to $-1$. There are $D \cdot (N - t)$ actions in $\mathcal{A}(s)$ that correspond to fill in one of the empty slots by one integer in $\{0, \ldots, N-1\}$. The generative process will not stop until sequences are fulfilled. By definition, $\mathcal{G}$ is a graded DAG and $\mathcal{S}_N = \mathcal{X} = \{0, \ldots, N\}^D$. We use the reward values provided in the dataset directly. Following Shen et al. (2023), we use reward exponents of 3 and 5, and normalize rewards to $[0.001, 10]$ and $[0.001, 100]$ for the SIX6 and QM9 datasets respectively. The guided distribution design also follows Shen et al. (2023).

**Model Architecture** Policies are constructed in the same way as the hyper-grid modeling experiment.

To further validate the relationships between policy-based methods and TB-based methods as explained in Appendix B.4. We compare the performance of TB-U, TB-Q and RL-Q. TB-Q denotes training GFlowNets by TB-U with trajectory sampler $P_\mathcal{D} = P_F$. RL-Q denotes training GFlowNets by RL-U with $Q$ and $V$ estimated empirically from the training data batch $\mathcal{D}$ during each training iteration. As shown in Fig. 10, the training behaviors of TB-Q and RL-Q are quite close. This validates our claims that when $P_\mathcal{D} = P_F$, TB-based methods correspond to represent $Q$ and $V$ empirically and our policy-based methods represented them functionally providing more robust gradient estimation.

## D.3 BAYESIAN NETWORK STRUCTURE LEARNING

**Environment** A Bayesian Network is a probabilistic model that represents the joint distribution of $N$ random variable and the joint distribution factorizes according to the network structure $x$:

$$P(y_1, \ldots, y_N) = \prod_{n=1}^{N} P(y_n | Pa_x(y_n)) \tag{78}$$

|        | SIX6 | QM9 |
|--------|------|-----|
| Method | $D_{TV}$ | $D_{TV}$ |
| DB-U | $0.175 \pm 0.030$ | $0.139 \pm 0.035$ |
| TB-U | $0.203 \pm 0.011$ | $0.056 \pm 0.007$ |
| RL-U | $0.167 \pm 0.008$ | $0.042 \pm 0.004$ |
| RL-T | $0.168 \pm 0.014$ | $0.057 \pm 0.007$ |
| DB-B | $\mathbf{0.154} \pm 0.060$ | $0.147 \pm 0.041$ |
| TB-B | $0.202 \pm 0.015$ | $0.070 \pm 0.007$ |
| RL-B | $0.166 \pm 0.011$ | $0.035 \pm 0.005$ |
| RL-G | $0.164 \pm 0.011$ | $\mathbf{0.035} \pm 0.006$ |

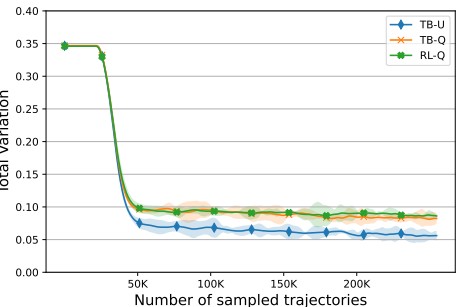

Figure 9: Sequence design performance comparison of the GFlowNets trained by different strategies.

Figure 10: Peformance comparison between TB-U and TB-Q and RL-Q

where $Pa_x(y_n)$ denote the set of parent nodes of $y_n$ according to graph $x$. As the structure of any graph can be represented by its adjacency matrix, the state space can be defined as $\mathcal{S} := \{s|\mathcal{H}(s) = 0\} \in \{0,1\}^{N \times N}$ where $\mathcal{H}$ corresponds to the acyclic graph constraint (Deleu et al., 2022), the initial state $s^0 = \{0\}^{N \times N}$ and specially $s^f := \{-1\}^{N \times N}$ in our implementation. For each state $s$, $a \in \mathcal{A}(s)$ can be any action that turns one of 0 values of $s$ to be 1 (i.e. adding an edge) while keeping $\mathcal{H}(s') = 0$ for the resulting graph $s'$, or equal to $(s \to s^f)$ that stopping the generative process and return $x = s$ as the terminating state. By definition, the corresponding $\mathcal{G}$ is not a graded DAG.

Given observation dataset $\mathcal{D}_y$ of $y_{1:N}$, the structure learning problem can be understood as approximating $P(x|\mathcal{D}_y) \propto P(x, \mathcal{D}_y) = P(\mathcal{D}_y|x)P(x)$. Without additional information about graph structure $x$, $P(x)$ is often assumed to be uniform. Thus, $P(x|\mathcal{D}_y) \propto P(\mathcal{D}_y|x)$ and the reward function is defined as $R(x) \propto P(\mathcal{D}_y|x)$. Distribution $P(\mathcal{D}_y|x)$ is also called graph score and we use $BGe$ score (Kuipers et al., 2014) in our experiment. Following Deleu et al. (2022), the ground-truth graph structure is generated from Erdős–Rényi model, and the observation dataset $\mathcal{D}_y$ with 100 samples is simulated from the ground-truth graph. The guided distribution design follows the hyper-grid experiment. A low probability value, $10^{-5}$ is assigned to the probability of terminating at states with $logR_{max}(\cdot) - logR(\cdot) < 10$.

**Model Architecture** Policies are constructed in the same way as the hyper-grid modeling experiment, but adjacency matrices are fed into neural networks directly without encoding.

| Method | $D_{TV}(\times 10^{-2})$ | $D_{JSD}(\times 10^{-3})$ | Method | $D_{TV}(\times 10^{-2})$ | $D_{JSD}(\times 10^{-3})$ |
|--------|------|------|--------|------|------|
| DB-U | $5.26 \pm 1.12$ | $3.91 \pm 0.32$ | DB-B | $5.92 \pm 1.64$ | $3.39 \pm 0.12$ |
| TB-U | $2.95 \pm 0.26$ | $2.84 \pm 0.07$ | TB-B | $4.69 \pm 0.80$ | $5.22 \pm 1.23$ |
| RL-U | $3.52 \pm 2.03$ | $4.33 \pm 3.13$ | RL-B | $6.10 \pm 0.67$ | $6.31 \pm 1.08$ |
| RL-T | $\mathbf{2.35} \pm 0.13$ | $\mathbf{2.15} \pm 0.16$ | RL-G | $3.72 \pm 0.49$ | $4.60 \pm 1.61$ |

Table 2: BN structure learning performance comparison of the GFlowNets trained by different strategies.

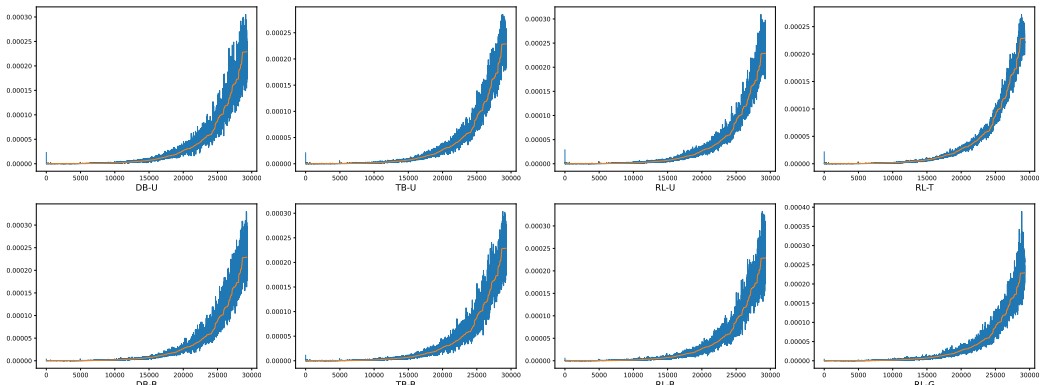

Figure 11: In each plot, the blue lines are the graphical illustrations of $P_F^\top(x)$ averaged across five runs of a training strategy for the BN structure learning experiment. The orange line is the ground-truth distribution and its values are plotted in an increasing order.

