# OpenReview forum: "GFLOWNET TRAINING BY POLICY GRADIENTS"
_ICLR.cc/2024/Conference — Submitted to ICLR 2024_

### Official Review · Reviewer_cxrP · 2023-10-30

**Soundness:** 3 good
**Presentation:** 2 fair
**Contribution:** 2 fair
**Rating:** 5
**Confidence:** 3

**Summary:**

This paper derives a connection between generative flow networks (GFlowNets) and policy gradient methods. An equivalence of gradients between GFlowNet objectives and gradients of variational objective is (re)derived, and from this emerges an equivalence with a reinforcement learning problem that can be solved using policy-based RL methods such as TRPO. Experiments are performed on three amortized sampling problems from past work and the algorithm is claimed to deliver improved samplers on two of them.

**Strengths:**

- The connection between GFlowNets and policy-based RL is stated for the first time, which can be valuable if it can produce improved training algorithms for GFlowNets.
- New result extending the SubTB gradient analysis from [Malkin et al., 2022b] to state flows.
- Experiments on diverse problems, representing a nontrivial engineering effort by the authors.

**Weaknesses:**

- There are many instances of unclear and incorrect mathematical language and notation in section 2 and 3.
  - Section 2.1:
    - "Assume there are $T+1$ topological orderings" -- that is not what "topological ordering" means; a topological ordering is a partial order homomorphism from a given partial order to a total order (i.e., a list of the states where no state is preceded by its descendant). Is it meant that the DAG is graded (as stated in the following paragraph) and that the layers are indexed by $0,\dots,T$?
    - Similarly, an instance of two elements being in a binary relation is not a "partial order". A partial order is a set with a binary relation (satisfying some properties), equivalent to a DAG.
    - It is incorrect to **define** ${\cal S}_0:=\{s_0\}$ and similar for ${\cal S}_T$, if the stratification into ${\cal S}_t$ already comes with the data of a graded DAG. That $s^0$ is the only state in the first graded component ${\cal S}_0$ is an **assumption/requirement**.
  - Section 2.2:
    - The second sentence of 2.2 is self-contradictory. "Probability measure" means the measure of the entire $\cal T$ is 1. In fact $F$ is just a measure, and $P$, its normalization, is a probability measure.
    - Equation (2) is missing a condition that this should hold only for $s\neq s^0$.
    - In equation (3), backward policy was not introduced/defined.
    - I did not understand this: "$\hat Z$ is a constant whose value is clamped to $Z$".
  - Section 3 start: I don't believe [Madan et al., 2023] discusses the relationship between TB and KL divergence. However, that relationship is discussed in [Malkin et al., 2022b] and in [Zimmermann et al., "A variational perspective on generative flow networks", TMLR].
  - Section 3.1:
    - "Expectation over $P_B$ is intractable due to unknown $Z^*$" -- even with known $Z^*$, sampling $P_B$ is not necessarily tractable.
    - "This forward gradient equivalence [in [Malkin et al., 2022b]] does not take the total flow estimator into account" -- in fact, that paper shows that the equivalence on $P_F$ parameter gradients holds independently of $Z$ and *also* states the equivalence for the $\log Z$ gradient (after equation (12) in that paper). Thus there is **nothing original in Proposition 1**.
    - In Proposition 1, it should be made clear that we do not propagate gradients to the distribution $P_F$ over which the expectation is taken. Otherwise, the proposition is not true. (This remark is made in Appendix A but not pointed to.)
- The paragraphs following Definition 1 are extremely difficult to understand, switching between GFlowNets and regular RL settings.
  - I object to the statement that in "regular RL" the Markov chain determined by the policy and transition environment is ergodic (many RL settings feature "irreversible" actions).
  - It should be noted that the math following Definition 1 is related to that in max-ent RL, which effectively places negative log-policy into the reward.
    - Most importantly, how can the policy gradient methods described here work off policy (which is the important advantage of GFlowNets, and important for preventing mode collapse)?
- The second sentence of section 3.3 seems key but does not make sense to me. The backward policy is a distribution over parents of each state and induces a distribution over subtrajectories. How can the $P_B$ be higher/lower for trajectories that precede a high-reward state $x$ if it is a distribution conditioned on $x$?
  - Also, please check grammar in that sentence.
- The experiment results are not very convincing for a few reasons:
  - On the hypergrid, the policy-based methods effectively learn from partial trajectory information, so a fairer comparison may be to subtrajectory balance (which actually performs better on large hypergrids [Madan et al.]).
  - Why is RL-G not tested in sections 4.2 and 4.3?
  - How were training hyperparameters selected?
  - I am not convinced by the evaluation of total variation in 4.2 and 4.3 if it is performed by sampling, which can induce high variance in the estimate. It should be possible to compute the true terminating distribution, at least for the structure learning problem. JSD between the full distributions would allow comparison with past work; right now, there is no way to validate that TB faithfully reproduces results from [Deleu et al.] and [Malkin et al., 2022b].
  - The same concerns apply to the bit sequence problem (4.2).
- Miscellaneous:
  - Please check your citation type (`\citet`/`\citep`) and consistent capitalization in "GFlowNet", which is wrong in multiple places.
  - Broken ref at the bottom of p.5.

**Questions:**

Please see above.

---

> ### Author Response · Authors · 2023-11-20
> **Response (Part I)**
>
> ## Weakness
> 1. Unclear and incorrect mathematical language and notation in section 2 and 3:
>    * Section 2.1:
>       * Yes, we meant that the DAG is graded and that the layers are indexed by $0,\ldots,T$. We have rephrased the sentence as ''Assuming that there is a topological ordering $S_1,\ldots,S_T$ for $T$ disjoint subsets of $S$''. The definition of the topological ordering follows Koller & Friedman (2009)[1].
>       *  We follow definition 1 of  Bengio et al. (2021b)[2], where a partial order, is a homogeneous relation on a set that is reflexive, antisymmetric, and transitive (Wallis,2011)[3].
>       * We have replaced the definitions as requirements.
>    * Section 2.2:
>       * We correct this as ''$F$ is a measure''.
>       * We add the missing condition in equation (2).
>       * We add the definition of the backward policy below equation (1).
>       * In terms of distributions, $\hat{Z}$ is the normalizing constant. Supposing a categorical distribution $P(\cdot)$ for $K$ kinds of states with parameters $(\theta_0,\ldots,\theta_k,\ldots,\theta_K)$, $P(\cdot)$ should be defined as $P(s^k):=\frac{\theta_k}{\hat{Z}}$ with $\hat{Z}=\sum_{k=0}^K\theta_k$, so that $\sum_{k=0}^K P(s^k)=1$. In terms of gradient computation, $\hat{Z}$ is considered constant w.r.t $\theta_{1:K}$, so $\nabla P(s^k|\theta_{0:K})=\frac{\nabla \theta_k}{\hat{Z}}$ . Distribution $\mu(s^0)=\frac{Z}{\hat{Z}}$ corresponds to a categorical distribution with only one category and one parameter $Z$. Thus, $\hat{Z}=Z$, which we call ''$\hat{Z}$ is clamped to $Z$". In terms of the gradient computation, $\nabla\mu(s^0;Z)=\frac{\nabla Z}{\hat{Z}}$.
>    * The beginning of Section 3:
>       * We thank the reviewer for pointing out the wrong citation, which has been corrected. Zimmermann et al. (2022)[4] do talk about the relationship between TB and KL. Their main contributions are KL-based objectives, which are shown experimentally to be effective objectives for GFN training. Nevertheless, the authors do not establish the relationship between TB and KL. So we put the citation to this paper in the `Related Work' Section.
>    * Section 3.1:
>       *  We have rephrased the sentence as "the backward gradient equivalence requires computing the expectation over $P_B^\top(x)=R(x)/Z^*$, which is impossible". What we mean here is that: to build a practical training method following the equivalence by Malkin et al. (2022)[5], we need to do sampling from $P_B^\top(x)=R(x)/Z^\ast$. Sampling from $R(x)/Z^\ast$ is impossible by our assumption and this is just what GFlowNet is designed for. However, sampling from $P_B(\tau|x)$ conditioning on $x$ is tractable since $P_B(\tau|x)=\prod_{t=1}^{T}P_B(s_{t-1}|s_{t})$ ($s_T=x$) and $P_B(\cdot|\cdot)$ is the backward sampling policy. Thus our proposition 1 can imply a practical training method over $P(B|x)\rho(x)$.
>       * We acknowledge that Malkin et al. (2022)[5] did consider gradients w.r.t $b:=logZ$ but did not establish the equivalence between TB and KL w.r.t $b$ in equations (7) and (8) of this paper.  Equation (12) does not complete the equivalence. It shows that: when you compute the gradients of the TB objective w.r.t $b$,  $\nabla_{b}L_{TB}(b)=E_{P_F(\tau)}\left[\nabla_{b}\left(c(\tau)+b\right)^2\right]=2E_{P_F(\tau)}\left[\left(c(\tau)+b\right)\right]$ ($c(\tau):=log\frac{P_F(\tau)}{R(x)P_B(\tau|x)}$); then
> gradient-based updating rules of $b$ with the learning rate $\frac{\eta}{2}$ is
> $b'\leftarrow
>  b-\eta E_{P_F(\tau)}\left[\left(c(\tau)+b\right)\right]=(1-\eta)b-\eta E_{P_F(\tau)}\left[c(\tau)\right]=(1-\eta)b+\eta b_{local}$, which is the equation (12).  We can see that no equivalence between TB and KL w.r.t $logZ$ was established during the whole derivation.  In light of this and our answer above about "sampling from $P_B$', we disagree with the comments that `nothing original in Proposition 1".
>       * We have moved the remark below Proposition 1 in the revised manuscript.
>
> [1]: Daphne Koller and Nir Friedman. Probabilistic graphical models: principles and techniques. MIT
> press, 2009.
>
> [2]: Yoshua Bengio, Salem Lahlou, Tristan Deleu, Edward J Hu, Mo Tiwari, and Emmanuel Bengio.
> GFlowNet foundations. arXiv preprint arXiv:2111.09266, 2021b.
>
> [3]: Walter Denis Wallis. A beginner’s guide to discrete mathematics. Springer Science & Business
> Media, 2011.
>
> [4]: Heiko Zimmermann, Fredrik Lindsten, Jan-Willem van de Meent, and Christian A Naesseth. A
> variational perspective on generative flow networks. arXiv preprint arXiv:2210.07992, 2022.
>
> [5]: Nikolay Malkin, Salem Lahlou, Tristan Deleu, Xu Ji, Edward Hu, Katie Everett, Dinghuai Zhang,
> and Yoshua Bengio. GFlowNets and variational inference. arXiv preprint arXiv:2210.00580,
> 2022.

---

> ### Author Response · Authors · 2023-11-20
> **Response (Part II)**
>
> 2. The paragraphs following Definition 1 are extremely difficult to understand:
>     * We have significantly revised the text. We acknowledge that the MDPs are either ergodic or absorbing.  What we mainly convey here is that: under the absorbing MDP specified by a DAG, functions $V$ and $Q$ can be time-invariant; by contrast, they have to be time-varying in general cases since the graph formed by all possible paths can contain cycles.
>     * In Section 7.2 of Bengio et al. (2021b)[2], the authors talked about the relationship between
>     Maximum Entropy (MaxEnt) RL and GFlowNets. The main message is that MaxEnt RL can be understood as learning a policy such that the terminating state distribution $P^\top(x)\propto n(x)R(x)$, where $n(x)$ is the number of paths in the DAG of all trajectories that lead to $x$, which is fundamentally different from the task of GFlowNets to learn a policy such that $P^\top(x)\propto R(x)$. This fact is also the main motivation of the GFlowNet model as explained in Proposition 1 of Bengio et al. (2021a)[6].
>         *  Policy-based RL can be run in off-policy ways (e.g. by importance sampling (Degris et al., 2012)[7]) and can also utilize temperature conditioning, but it is beyond the scope of our work.
> 3. The second sentence of Section 3.3  seems key but does not make sense:
>     * We suggest referring to the sections "Compositional reward functions" and "The substructure credit assignment problem" by Shen et al. (2023)[8]. We are not conditioning on $x$ for sub-trajectories. First, $P_B(\cdot|\cdot)$ is the backward policy, and by definition, the backward distribution of a sub-trajectory $\bar{\tau}$ that ends in $s_n$ (i.e conditioning on $s_n$) is $P_B(\bar{\tau}=(s_m\rightarrow \ldots\rightarrow s_n)|s_n)=\prod_{t=m+1}^n P_B(s_{t-1}|s_{t})$.  Now let's suppose that $\bar{\tau}\in \tau|x=(s^0\rightarrow\ldots s_m\rightarrow \ldots s_n\rightarrow,\ldots x\rightarrow s^f)$ and $R(x)$ is high. Now if we want the probability of sampling $\tau$ that ends in $x$ to be high, we need to make the probability of sampling $\bar{\tau}$ to be high.
>         *   We have corrected the grammar error.
> 4. The experiment results are not very convincing:
>      * Yes, sub-trajectory objectives may help improve performance. But how to properly split a complete trajectory in the multiple sub-trajectories and how to combine the responding sub-trajectory objectives so that the overall performance can be improved is a tricky engineering problem. As our work focuses on connecting GFlowNet training to RL,  we do not consider sub-trajectory distribution following the choices of Malkin et al. (2022)[5] and Shen et al. (2023)[8], but our RL methods do apply to sub-trajectories as shown in Proposition 3 in Appendix A.3 and we provide additional results under the Detailed Balance Objective. (The detailed Balance objective corresponds to sub-trajectories with length 1, and TB corresponds to complete trajectories. )
>     * We add the missing results under RL-G for other experiments.
>     * We add the detail about the hyper-parameter setup in the appendix.
>     * We use dynamical programming to compute the learned terminating distribution and update the results when the space scale is relatively small. We add the performance results measured by JSD.
> 5. Miscellaneous:
>     * We thank the reviewer for pointing out these typos. We have corrected the citation typos and capitalization in ``GFlowNet".
>     * We have fixed the broken reference.
>
>
> [2]: Yoshua Bengio, Salem Lahlou, Tristan Deleu, Edward J Hu, Mo Tiwari, and Emmanuel Bengio.
> GFlowNet foundations. arXiv preprint arXiv:2111.09266, 2021b.
>
>
> [5]: Nikolay Malkin, Salem Lahlou, Tristan Deleu, Xu Ji, Edward Hu, Katie Everett, Dinghuai Zhang,
> and Yoshua Bengio. GFlowNets and variational inference. arXiv preprint arXiv:2210.00580,
> 2022.
>
> [6]: Emmanuel Bengio, Moksh Jain, Maksym Korablyov, Doina Precup, and Yoshua Bengio. Flow
> network based generative models for non-iterative diverse candidate generation. Advances in
> Neural Information Processing Systems, 34:27381–27394, 2021a.
>
> [7]: Thomas Degris, Martha White, and Richard S Sutton. Off-policy actor-critic. arXiv preprint
> arXiv:1205.4839, 2012.
>
> [8]: Max W Shen, Emmanuel Bengio, Ehsan Hajiramezanali, Andreas Loukas, Kyunghyun Cho, and
> Tommaso Biancalani. Towards understanding and improving GFlowNet training. arXiv preprint
> arXiv:2305.07170, 2023.

---

> ### Comment · Reviewer_cxrP · 2023-11-20
> **Response**
>
> Thanks for the clarifications and improvements.
>
> I am still having trouble with the second sentence of 3.3. If $x$ has high reward, you want the **forward** sampling probability of trajectories leading to $x$ to be high. That has nothing to do with the backward probability, conditioned on $x$. Possibly I am missing something. What would the statement mean in the case of a tree-structured MDP (so $P_B$ is always 1)?
>
> Another minor comment: please check your citations. They are often to the arXiv version when the paper was in fact published in a journal or conference proceedings, in particular: [Bengio et al, 2021b] -- JMLR, [Kingma and Welling, 2013] -- ICLR 2014, [Malkin et al., 2022a] -- NeurIPS 2022, [Malkin et al., 2022b] -- ICLR 2023, [Schulman et al., 2015b] -- ICLR 2016, [Zimmermann et al., 2022] -- TMLR.

---

> ### Author Response · Authors · 2023-11-21
> **Response to reviewer cxrP**
>
> * During GFlowNet training $P_B(\tau)$ serves as one component of the objective, as $P_B(\tau|x)R(x)$ specifies the form of the desired flow $F^\ast(\tau)$ such that $\sum_{x\in \tau}F^\ast(\tau)=F^\ast(x)=R(x)$. Accordingly, $P_F$ also specifies a flow $F(\tau)= P_F(\tau|s_0)Z$. The training task is to minimize the mismatch of $F(\tau)$ and $F^\ast(\tau)$ so that the mismatch of $F(x)$ and $F^\ast(x)$ is also minimized. Now Let $\tau_{\leq n}$ denote the sub-trajectory of some $\tau$ with high reward $R(x)$. Then, if the induced sub-trajectory flow implied by $P_B$,  $F^\ast(\tau_{\leq n})=P_B(\tau_{\leq n}|s_n)F^\ast(s_n)$ is high, the sub-trajectory flow $F(\tau_{\leq n})= P_F(\tau_{\leq n}|s_0)Z$ will also be required to be high.  Since trajectories are sampled by $P_F(\tau)$ via $F(\tau)$, the chance that high-rewarding $\tau$ is sampled by $P_F(\tau)$ is increased.  Besides, the probability of $P_F(\tau_{\leq n})$ is usually higher than $P_F(\tau)$ as $\tau_{\leq n}$ can be not only the sub-trajectory of some high rewarding $\tau$ but also some low rewarding $\tau$. Finally, as the mismatch between $F^\ast(\tau)$ and $F(\tau)$ mainly involves these high-rewarding trajectories, the training efficiency can be improved in this way.
>
>     *   We acknowledge that there can not be any guidance from $P_B$ in tree-MDPs where each state has only one parent and one backward action accordingly. Actually, this is also the reason why we did not follow exactly Shen et al. (2023) when doing biological sequence design. Shen et al. (2023) defined an Append/Prepend MDP environment, which is not a tree-MDP but each state has two/one parents and two/one backward actions accordingly. We design a more flexible MDP environment, where $s_t$ has $t$ backward actions.
>
> * Thanks again for pointing out this. We have corrected these citations.

---

### Official Review · Reviewer_gXTK · 2023-10-30

**Soundness:** 3 good
**Presentation:** 2 fair
**Contribution:** 2 fair
**Rating:** 5
**Confidence:** 4

**Summary:**

This work shows that there is an equivalence (in expectation) between training GFlowNets (GFN) with the Trajectory Balance objective, and a special form of non-stationary-reward policy gradient (PG) method. This is derived through the equivalence (in expectation) between TB and a forward KL-Divergence shown in prior work. This enables the algorithm to be treated as a PG method, onto which one can apply methods like TRPO.

The authors perform some empirical evaluations of their method on grids, bit sequences, and Bayesian networks.

**Strengths:**

This paper continues the work of relating GFlowNets and other frameworks, here RL, and of building novel methods upon that. In that sense the work here is a novel and interesting contribution.

**Weaknesses:**

The two main weaknesses of this paper are presentation and strength of empirical results.

Presentation: what the authors are trying to do isn't incomprehensible, I kind of get it and staring at the appendix helps, but I honestly think  section 3 could be heavily reworked to be more concise and tell a story, e.g. make it clear at every step what the symbols are, where they come from and how they make us progress in going from TB to RL with a non-stationary reward.

Results:
- Hypergrids and bit sequences are not very hard nor interesting problems, they were meant to be toy examples in the papers in which they were introduced. Learning to produce Bayesian nets is interesting, but I don't think the authors properly compare to Deleu et al.
- There isn't much in terms of "why does this work [better]?" which is usually a good thing to have in a paper.


Now I realize this paper may be more "theory" than experiment, but it does feel unfortunate that there's a mismatch between the "we're doing PG therefore we can bring in the whole RL arsenal" and the lack of big results.

This may be tangential, and just a semantics argument, but I find it slightly far-fetched to call this a policy gradient algorithm, or maybe even RL. The reason: the reward changes at every state, _and_ in a self-referential manner. The closest we have is maybe methods like Random Network Distillation, but even then the update of $R$ is decoupled from the update of $\pi$. Just because the update equation "looks like" PG, doesn't mean it's the most "scientifically useful" way to describe it. Methods like TRPO certainly assume a fixed reward, so I don't even know if what's being done here is correct.

**Questions:**

- > "Value-based methods, however, face the difficulties of designing a powerful sampler that
can balance the exploration and exploitation trade-off, especially when the combinatorial space is
enormous with well-isolated modes."

  I really don't see what this challenge has to do with a method being value-based or not. Soft-Q-Learning is very powerful and handles this trade-off. SubTB-GFlowNets seem to work very well and handle this trade-off through means like temperature conditioning.
- I don't understand if the authors mean something specific (or different) when they talk about absorbing MDPs; GFlowNets already operate on absorbing DAGs through the definition of $s_f$.
- In section 3.2, the authors write "It is clear that...", I beg to differ, it would be helpful to explain where this equation is pulled from. Why can $V_F$ be substituted for (4)?
- The authors carry on to write that the gradients of TB can be written something like $(Q(s,a) -C)\nabla \log P_F$, which sounds cool but at this point in the paper $Q$ (nor $V$) hasn't been fully defined. It appears to be the action-value function of policy $P_F$, but for which reward? For which MDP?
- "Following the pipeline by Shen et al. (2023), we must solve the optimization of LT B−G to find the desired PB at first". I'm not 100% sure but fairly confident that in Shen et al. $P_B$ and $P_F$ are "trained" simultaneously, it's just that $P_B$ is non-parametric, thus it evolves in time as more states are visited.
- Hypergrid:
    - Gridworlds, in GFNs and in RL, are useful to _sanity check_ an idea, but I really wouldn't use it as a strong base of comparison between algorithms
    -  "[P] can be estimated empirically by sampling" in a hypergrid it can be computed exactly as well through dynamic programming.
- BN: Deleu et al. use detailed balance, not TB, to train their model. It would be good to compare performance to their setup exactly rather than a possibly suboptimal one.
- After Eq (12), I'm not sure what Lemma 1 has to do with the fact that the expected value of a score is 0.


- There are lots of typos and some citation formatting mistakes throughout, I'd recommend another round of proofreading.

---

> ### Author Response · Authors · 2023-11-20
> **Response (Part I)**
>
> ## Weakness
> *  Presentation:
>      * We rewrite Section 3 as suggested to make every step of derivations clearer.
> *  Results:
>     * We re-conduct the Bayesian Network structure learning experiment with additional results under the Detailed Balance objectives.
>     * We have added more explanations about `why it is good' in the experiment section.
> * We understand your concerns that some people bring in unnecessary tools to solve problems that are trivial or have been solved. However, Bengio et al. (2021a)[1] do not actually consider the problems in reinforcement learning and we bridge their work to reinforcement learning by defining the proper value function and other RL components, and consequently to further improve the performances.  As shown in the hyper-grid experiment, our new RL-formulated method can solve the problem not only more quickly (convergence rate), but also much more scalable to a larger scale than being reported in Bengio et al. (2021a)[1] and Malkin et al. (2022)[2]. Therefore, We think our work can serve as a stepping stone to make GFlowNets useful in real-world applications. Besides,  we redesigned the bit-sequence experiment as bio-sequence experiments based on real-world datasets to further validate our proposed methods.
>
> * Yes, we agree that TRPO assumes that the reward is fixed.  In light of this fact, we introduce Theorem 2 as the theoretical guarantee with the non-stationary reward and absorbing MDP.  Besides, we add more explanations in Appendix B.4 about the stability of our methods under non-stationary rewards.
>
> [1]: Emmanuel Bengio, Moksh Jain, Maksym Korablyov, Doina Precup, and Yoshua Bengio. Flow
> network based generative models for non-iterative diverse candidate generation. Advances in
> Neural Information Processing Systems, 34:27381–27394, 2021a.
>
> [2]: Nikolay Malkin, Salem Lahlou, Tristan Deleu, Xu Ji, Edward Hu, Katie Everett, Dinghuai Zhang,
> and Yoshua Bengio. GFlowNets and variational inference. arXiv preprint arXiv:2210.00580,
> 2022.

---

> > ### Comment · Reviewer_gXTK · 2023-11-20
> >
> > Thanks for the update and precisions. I'd like to note that the current revised paper exceeds the [9 page limit](https://iclr.cc/Conferences/2024/CallForPapers) that will be enforced as well for the camera ready.
> >
> > > As shown in the hyper-grid experiment, our new RL-formulated method can solve the problem not only more quickly (convergence rate), but also much more scalable
> >
> > Right, but my point is that solving the hypergrid faster is not necessarily predictive of solving general problems faster. Thus my recommendation to reduce the time dedicated to that problem in the paper. Biological sequence & BN experiments are more interesting in that regards.
> >
> > > We have added more explanations about `why it is good' in the experiment section.
> >
> > My comment might not have been clear enough. All empirical work in this paper is of the form "we tried X and performance was Y". It's nice to have hypotheses in the text that link X to Y, but it's much nicer to test these hypotheses with data. Let me give an example, if someone introduced a weight decay in DNNs that penalize activations that are greater than $\pi/4$ then I'd expect to see two plots, one showing that the proposed weight decay indeed reduces the number of such activations, and another showing some correlation or link between the number of activations greater than $\pi/4$ and performance/memorization/learning speed, or such.
> >
> > Other comment, the QM9 task is not a biological sequence design task, but a material design task. Jain et al first used it in a GFN setting, introducing it as an atom-graph generation problem. Shen et al introduced a simplified variant based on strings which induces a much smaller state space.

---

> ### Author Response · Authors · 2023-11-20
> **Response (Part II)**
>
> ## Question
> * Yes, for regular RL problems, soft-Q learning can balance the trade-off.  However, soft-Q learning relies on the extended Bellman equation that incorporates a KL divergence as the regularized term. In terms of GFlowNets, it means that a KL regularized term should be incorporated into the flow balance equation, which is not straightforward. We do acknowledge that it is a good research direction. Actually, in Section 7.2 of Bengio et al. (2021b)[3], the authors talked about the relationship between GFlowNet and Maximization Entropy (MaxEnt) RL, and soft-Q learning is one of the representative MaxEnt RL methods. It pointed out that MaxEnt RL can be understood as learning a policy such that the terminating state distribution $P^{\top}(x)\propto n(x)R(x)$, where $n(x)$ is the number of paths in the DAG of all trajectories that lead to $x$, which is different from the task in GFlowNets to learn a policy such that $P^{\top}(x)\propto R(x)$. This fact is also the main motivation of the GFlowNet model as explained in Proposition 1 of Bengio et al. (2021a)[1]. By contrast, our work formally bridges the "flow" in GFlowNets to RL by defining non-stationary rewards, allowing us to perform the same tasks as GFlowNet in RL formulations.
>     * Sub-trajectory objectives may help improve performance. But how to properly split a complete trajectory in the multiple sub-trajectories and how to combine the responding sub-trajectory objectives so that the overall performance can be improved is a tricky engineering problem. As our work focuses on connecting GFlowNet training to RL, we do not consider sub-trajectory distribution following the choices of Malkin et al. (2022)[2] and Shen et al. (2023)[4], but our RL methods do apply to sub-trajectories as shown in Proposition 3 in Appendix A.3. Besides, temperature conditioning corresponds to using an off-policy sampler, which is different from the forward policy during training. Policy-based RL can be run in off-policy ways (e.g. by importance sampling (Degris et al., 2012)[5]) and can also utilize temperature conditioning, but it is beyond the scope of our work.
>
> * We correct the statement as $G$ already specifies an absorbing MDP. What we want to convey here is that for a normal absorbing MDP, functions $V$ and $Q$ have to be time-dependent in that the graph formed by all possible paths can contain cycles. Here we want to emphasize the fact that when the graph contains no cycle, functions $V$ and $Q$ can be time-independent, giving rise to modeling convenience.
> * We rewrite Section 3 and add explanations in Appendix B.4 further clarifying Why $V_F$ can be substituted for (4).
> * We provided further explanations in Appendix B.4.
> * In the main text after Theorem 6.1 in Shen et al. (2023)[4], the authors stated that ''We propose to first train PB to match $p(\tau|x)$'' and ''Once converged, we freeze PB, then learn PF with fixed PB using trajectory balance, which recovers a proper GFlowNet learning objective''. Thus, in principle, their framework should be done in two phases. We acknowledge that in the following `Training Consideration' section, the authors mixed $P_B$ and $P_G$ by $\alpha P_B+(1-\alpha)P_G$ in the training objective w.r.t $P_F$, avoiding doing two-phase training in practice. This operation, however, lacks theoretical guarantees as the mixed distribution is still non-Markovian. By comparison, we provide a theoretical guarantee of our coupled training strategy.
> * Hypergrid:
>    * Following Shen et al. (2023)[4], we resign the bit-sequence experiment as bio-sequence experiments based on real-world datasets (including nucleotide strings and molecule graphs).
>    * We update the performance results computed by dynamic programming in the hyper-grid experiment.
> * we add results under the DB objective for all experiments.
> * $E_{P_{F,\mu}(\tau;\theta_F)}[\nabla_{\theta_F}c(\tau;\theta_F)]=0$ does not imply that the expected score $E_{P_{F,\mu}(\tau;\theta_F)}[c(\tau;\theta_F)]$ is zero.  Actually, $E_{P_{F,\mu}(\tau;\theta_F)}[\nabla_{\theta_F}c(\tau;\theta_F)]=E_{P_{F,\mu}(\tau;\theta_F)}[1\cdot\nabla_{\theta_F}log P_F(\tau|s_0;\theta_F)]=\nabla_{\theta_F}E_{P_{F,\mu}(\tau;\theta_F)}[1]$ by lemma 1. Furthermore $\nabla_{\theta_F}E_{P_{F,\mu}(\tau;\theta_F)}[1]=\nabla_{\theta_F}1=0$.
> * We thank the reviewer to point out the typos and have corrected the typos.
>
> [3]: Yoshua Bengio, Salem Lahlou, Tristan Deleu, Edward J Hu, Mo Tiwari, and Emmanuel Bengio.
> GFlowNet foundations. arXiv preprint arXiv:2111.09266, 2021b.
>
> [4]: Max W Shen, Emmanuel Bengio, Ehsan Hajiramezanali, Andreas Loukas, Kyunghyun Cho, and
> Tommaso Biancalani. Towards understanding and improving GFlowNet training. arXiv preprint
> arXiv:2305.07170, 2023.
>
> [5]: Thomas Degris, Martha White, and Richard S Sutton. Off-policy actor-critic. arXiv preprint
> arXiv:1205.4839, 2012.

---

> ### Author Response · Authors · 2023-11-21
> **Response to Reviewer gXTK**
>
> * Thanks for pointing out this issue. We have reorganized the main text so that it meets the 9-page limit.
> * Thanks for the clarification. We agree with your opinion that  Biological sequence & BN experiments are more interesting.
>
> * Thanks for the clarification. As our claims involve policy gradient estimation, which can not be easily visualized directly, we compare the performance of TB-based methods with $P_D=P_F$, and policy-based methods with $Q$ and $V$ being  represented empirically. Based on our discussion of the connection between TB-based methods and policy-based methods in Appendix B.4, their performance should be close. Based on the experimental results (Fig.10 in the Appendix), they do show such expected trends, and therefore validate our claims that the policy-based methods with $Q$ and $V$ represented functionally provide more robust gradient estimation.
>
> * We have rephrased "biological sequence design"  as "Sequence design based on real-world datasets"

---

### Official Review · Reviewer_MZqG · 2023-10-31

**Soundness:** 2 fair
**Presentation:** 1 poor
**Contribution:** 2 fair
**Rating:** 5
**Confidence:** 4

**Summary:**

The paper proposes a new approach for improving the training of GFlowNets from the perspective of policy-based rewards and guided backward policy. The proposed method includes a coupled training strategy that jointly solves GFlowNet training and backward policy design, which aims to improve training efficiency. The method is tested a few benchmark including hypergrid, bit sequence generation and bayesian structure learing.

**Strengths:**

This paper tries to connect the field of RL and GFlowNets, which forms an interesting problem. The experimental part is well-explained.

**Weaknesses:**

The main concern for the paper is that the experimental evaluation part is too toy -- which focuses on some synthetic problems including hypergrid and bit sequence generation only, and the evaluation metric does not follow previous paper, and some of the claims are not well supported.

**Questions:**

> Our theoretical results are also accompanied by performing experiments in three application domains, hyper-grid modeling, Bit sequence modeling, and Bayesian Network (BN) structure learning.

Could the authors also consider molecule generation and biological sequence design, which are more practical and larger-scale tasks for validating the effectiveness of the approach?

> Definition 1 (Policy-dependent Rewards).

If the rewards are dependent on the policy, will it change during the course of training? Doesn't this introduce additional instability?

Regarding the experiments, why not following the standard metric for evaluating the algorithms, e.g., L1 error, the number of modes?

---

> ### Author Response · Authors · 2023-11-20
> **Response**
>
> ## Weakness
> * As suggested, we redesign the bit-sequence experiment as molecular generation and biological sequence design experiments.
> We now redesign the bit-sequence experiment as bio-sequence experiments based on real-world datasets(Shen et al.,2023) [1]. As for the measurements, we provide the answers below.
>
> ## Question
> * As we explained, we have redesigned the experiments as suggested.
> * We have added more explanations in Appendix B.4 about the stability under non-stationary rewards in our revised manuscript.
> *  For the expected $l_1$-dist over the ground-truth and learned terminating state distributions, $P^\ast(x)$ and $ P^\top_F(x)$, it is estimated by $\frac{1}{|\mathcal{X}|}\sum_x |P^\ast(x)-P^\top_F(x)|$ in previous works. By contrast, the total variance $D_{TV} $is computed by $\frac{1}{2}\sum_x |P^\ast(x)-P^\top_F(x)|$. We can see that mean $l_1$-dist is not a very appropriate choice as $|\mathcal{X}|$ is large ($>10^4$) and $\sum_x |P^\ast(x)-P^\top_F(x)|\leq 2$. It can be observed that the reported $l_1$-dist is in scale below $10^{-3}$ in all the previous works where hyper-grid experiments were considered. Besides, we update the results table with performances measured by Jensen–Shannon Divergence (JSD) as done in previous works. Counting the number of nodes is suitable for sequence design experiments where we have a code book of high-reward sequences and we can check how many generated trajectories end in sequences within the code book. Since it is not a common measurement over distributions, we do not take this measurement into account.
>
> [1]: Max W Shen, Emmanuel Bengio, Ehsan Hajiramezanali, Andreas Loukas, Kyunghyun Cho, and
> Tommaso Biancalani. Towards understanding and improving GFlowNet training. arXiv preprint
> arXiv:2305.07170, 2023.

---

> ### Author Response · Authors · 2023-11-21
> **Gentle Reminder**
>
> Dear Reviewer
>
> Here is a gentle reminder that the rebuttal session will end soon, so we wonder whether you have some comments on our responses.
>
> Thanks for your time.

---

### Official Review · Reviewer_U7wX · 2023-11-01

**Soundness:** 3 good
**Presentation:** 4 excellent
**Contribution:** 3 good
**Rating:** 5
**Confidence:** 2

**Summary:**

This paper reformulates the the GFlowNet training problem as a RL problem over a special MDP. A policy gradient method is proposed to do RL training in this setting. The authors also formulate the design of backward policies in GFlowNets as a RL problem and propose a coupled training strategy. Theoretical and empirical results are provided to validate and help understand the proposed method.

**Strengths:**

**originality**
- The main novelty of the paper is on formulating the GFlowNet problems as RL problems and allow the use of RL in these problems. The problem formulation and the training strategies proposed can be considered novel results.
- The theoretical results are interesting and can be considered novel

**quality**
- The paper is overall well-written, with only minor isuses

**clarity**
- The paper is quite clear

**significance**
- The new problem formulation and training strategies discussed in this paper allow RL to be used in GFlowNet training problems, together with theoretical and empirical results, can be a significant contribution.

**Weaknesses:**

Related work:
- It seems to me the related work can be improved, for example, what is the most relevant standard RL algorithm? And how is your proposed policy gradient method different in design? Additionally, GFlowNet and RL are discussed together in some other papers in the literature, such as the "GFlowNet Foundations" by Bengio et al. How is the analysis in your work related to these previous works?

Additional technical details:
- Would be nice to have more technical details, for example, how does your method compare to others in terms of computation efficiency? Will formulating into RL problems make training much slower.

Ablations:
- I think more ablation on the proposed method with different hyperparameters can help us understand better how it can be applied/tuned to different problems.

Other issues:
- End of page 5, a figure reference seems to be broken. Is this figure in the paper?

**Questions:**

- Can you elaborate on why is RL a good strategy to use for these problems and what unique advantages it can bring? Are there concrete problems where they have to be solved with RL and not other strategies?

---

> ### Author Response · Authors · 2023-11-20
> **Response**
>
> ## Weakness
> ### Related work:
>   * We have improved the related work section to include the most relevant RL methods. Besides, we have thoroughly gone through the paper by Bengio et al. (2021b)[1]. We believe that only Section 7.2 formally talks about the relationship between GFlowNet and RL.
>     1. The first point of this section is that the traditional RL method aims at finding the highest-rewarding trajectories that render the maximization of expected return $R$ by all the probability mass on the highest-return trajectories. GFlowNets, however, try to assign proper probabilities to all possible trajectories according to their terminating reward, so that the trajectories end in terminating state $x$ with the probability proportion to $R(x)$. This fact is also emphasized on page 33, before the beginning of Section 4.7, PPO, an RL method that had previously been used for generating molecular graphs, tends to focus on a single mode of the reward function.
>
>      2. The second point is that Maximization Entropy (MaxEnt) RL can be understood as learning a policy such that the terminating state distribution $P^{\top}(x)\propto n(x)R(x)$, where $n(x)$ is the number of paths in the DAG of all trajectories that lead to $x$.  As explained by the authors, existing RL methods perform different tasks from GFlowNets, which learn a policy such that $P^{\top}(x)\propto R(x)$. Thus, GFlowNets (Bengio et al., 2021a)[2] originally do not fit into the traditional reinforcement learning framework as it lacks the formal definition of RL components. By contrast, our work formally bridges the "flow" in GFlowNets to RL by defining non-stationary rewards, allowing us to perform the  tasks as GFlowNets but in RL formulations.
>
> ### Additional technical details:
>   * We add the training time reports for hyper-grid experiments to show the efficiency. Generally speaking, the time cost slightly increases using policy-based methods except RL-T.  RL-T is relatively time-consuming in that estimating the Hessian-inverse-vector product is slow.
> ### Ablations:
>   * We think the ablation study is used to understand the impact of different components or features of a model on its overall performance. As we did not propose new components in policy-based optimization, we only compared our work with different GFlowNet algorithms and did not conduct an ablation study. We would truly appreciate your advice on how we may conduct the ablation study if there are specific modular components in our method requiring more careful investigation.
>   * We have fixed the broken citation of the figure.
> ## Question
>
>   * Firstly, the workflow of existing methods for GFlowNet training follows the logic of value-based RL methods. Secondly, we added Appendix B.4 to further explain that when the sampler policy $P_D=P_F$, TB-based methods correspond to estimate the $Q$ function empirically and at best, do variance reduction by a baseline $C$ which is constant w.r.t $P_F$. By contrast, $Q$ is approximated functionally, and $V$ serves as a functional baseline for variance
> reduction in our policy-based method, which typically renders more robust gradient estimation (Schulman et al., 2015)[3]. Besides, the major advantage of TB-based training is allowing off-line training (i.e $P_D\neq P_F$), which is also allowable by policy-based RL methods (Degris et al., 2012)[4], for example, by importance sampling.
>
> [1]: Yoshua Bengio, Salem Lahlou, Tristan Deleu, Edward J Hu, Mo Tiwari, and Emmanuel Bengio.
> GFlowNet foundations. arXiv preprint arXiv:2111.09266, 2021b.
>
> [2]: Emmanuel Bengio, Moksh Jain, Maksym Korablyov, Doina Precup, and Yoshua Bengio. Flow
> network based generative models for non-iterative diverse candidate generation. Advances in
> Neural Information Processing Systems, 34:27381–27394, 2021a.
>
> [3]: John Schulman, Philipp Moritz, Sergey Levine, Michael Jordan, and Pieter Abbeel. High-dimensional
> continuous control using generalized advantage estimation. arXiv preprint
> arXiv:1506.02438, 2015.
>
> [4]: Thomas Degris, Martha White, and Richard S Sutton. Off-policy actor-critic. arXiv preprint
> arXiv:1205.4839, 2012.

---

> ### Author Response · Authors · 2023-11-21
> **Gentle Reminder**
>
> Dear Reviewer
>
> Here is a gentle reminder that the rebuttal session will end soon, so we wonder whether you have some comments on our responses.
>
> Thanks for your time.

---

### Official Review · Reviewer_xjL5 · 2023-11-01

**Soundness:** 3 good
**Presentation:** 3 good
**Contribution:** 3 good
**Rating:** 5
**Confidence:** 4

**Summary:**

This work introduces a way to train GFlowNets through policy gradients from the RL literature. The work proposes a coupled training strategy to train the forward and backward policies in GFlowNets.

**Strengths:**

1. This work discusses an interesting direction of training GFlowNets and using learning from RL policy-based methods to introduce a policy-based GFlowNet.
2. The discussion and perspective around backward trajectories is interesting and a useful way to improve GFlowNet training.
3. The gradient equivalence discussion and analysis is useful to understand the theoretical claims and some of the motivation behind this work.

**Weaknesses:**

1. The related work section could be made more exhaustive by adding the other GFlowNet losses and their references.
2. It will be useful to expand the number of environment configurations. For hypergrid, N=2 and N=3 are the only options used and using a higher value will help.
3. Previous work has analyzed number of states visited for hypergrid domain, which has not been included here.
4. Adding other GFlowNet based baselines, such as Detailed Balance, would be useful as it is an important and commonly used objective.
5. Some of the domains that were used in the previous work have been not included here. In order to corroborate that the proposed method can be used over different settings, including them would be helpful.
6. For bit sequence domain, including all bits used in the previous work would be beneficial for a fair comparison across methods.
7. Related work could be expanded and made more exhaustive, including the recent improvements in GFlowNets as well.

**Questions:**

The main concerns are about the coverage of the experiments section. If the authors could address and answer those concerns, stated in the weaknesses section, it will be useful for the contribution of the work.

---

> ### Author Response · Authors · 2023-11-20
> **Response**
>
> ## Weakness
>
> 1. In the related work section, we have added references to all GFlowNet losses mentioned in this paper.
>  2. Following Malkin et al. (2022)[1] who consider these cases with $N=2,3,4$, we conduct an additional experiment with $N=4$ and $H=32$.
>  3. During the training process, training data is sampled in a trajectory-wise manner for both the existing GFlowNet training methods and our RL methods. Thus, we believe training curves w.r.t the number of sampled trajectories is a better way to compare the convergence rate. For example, the DAG graphs for Hyper-grid experiments are not graded. The trajectory length varies across trajectory samples, and the number of sampled states varies in each training iteration accordingly.
>   4. We update the existing results by adding training results under the Detailed Balance objectives.
>   5. We redesign the bit-sequence design experiment as bio-sequence design experiments (including nucleotide strings and molecule graphs) based on real-world datasets in  Shen et al.,2023[2].
>   6. Please refer to our answer above to weakness 5.
>   7. We expand the related work section by adding Shen et al.,2023[2] and Zimmermann et al[3].
> (2022).. For other existing improvements, we would truly appreciate your kind help if any specific references can be provided.
>
> [1]: Nikolay Malkin, Salem Lahlou, Tristan Deleu, Xu Ji, Edward Hu, Katie Everett, Dinghuai Zhang,
> and Yoshua Bengio. GFlowNets and variational inference. arXiv preprint arXiv:2210.00580,
> 2022.
>
> [2]: Max W Shen, Emmanuel Bengio, Ehsan Hajiramezanali, Andreas Loukas, Kyunghyun Cho, and
> Tommaso Biancalani. Towards understanding and improving GFlowNet training. arXiv preprint
> arXiv:2305.07170, 2023.
>
> [3]: Heiko Zimmermann, Fredrik Lindsten, Jan-Willem van de Meent, and Christian A Naesseth. A
> variational perspective on generative flow networks. arXiv preprint arXiv:2210.07992, 2022.

---

> ### Author Response · Authors · 2023-11-21
> **Gentle Reminder**
>
> Dear Reviewer
>
> Here is a gentle reminder that the rebuttal session will end soon, so we wonder whether you have some comments on our responses.
>
> Thanks for your time.

---

> ### Comment · Reviewer_xjL5 · 2023-11-22
> **Thank you for your responses and updates.**
>
> Thank you for taking time to provide clarifications and for adding revisions to the work, including comments from other reviewers.
> I am updating my scores and will update them further after proof-reading the document again to verify if overall all concerns have been addressed.

---

> > ### Author Response · Authors · 2023-11-23
> > **Response**
> >
> > Thank you very much for your updated comments and the time you have already invested in reviewing our work.

---

### Author Response · Authors · 2023-11-20
**Overall Response**

Dear Reviewers,

Thank you very much for your encouraging comments and constructive critiques. We have tried our best to address the your concerns and questions and modify the manuscript according to your suggestions. In particular,

1. We rewrote the main text (especially Section 3) to improve the presentation and added Appendix B.4 to further explain the relationships between TB-based and policy-based training methods.

2.  We add a hyper-grid experiment with $N=4$ and $H=32$.

3. As suggested, we redesigned the bit-sequence experiment as a biological sequence experiment based on real-world datasets, including nucleotide string and molecule graph generation.

4. We reconducted the Bayesian structure learning experiment with more concrete experiment results.

5. We add the results under the Detailed Balance (DB) objective for all the experiments and the missing results for RL-G.

We have uploaded an updated manuscript with highlights indicating the changes.  As a result of the revision, we believe that the overall quality of the paper has been further improved, and we again thank the you for the constructive remarks. Please note that we have highlighted the newly added or significantly revised sections’ titles, figures, and tables’ captions, and other major changes in the revised manuscript using blue highlighting (minor corrections have not been highlighted).

Sincerely.

---

> ### Author Response · Authors · 2023-11-22
> **Gentle Reminder**
>
> Dear Reviewer xjL5, U7wX and MZqG
>
> As the rebuttal period is approaching its end, we would be grateful if you could take a moment to review our responses. Your further insights will be immensely beneficial for us to improve our work and to reach a well-informed decision.
> We understand that reviewing is a demanding task and truly appreciate the time and effort you have already invested in reviewing our work. In response to your valuable feedback, we have provided a detailed rebuttal addressing the questions and concerns raised.
> Please feel free to reach out if there are any additional clarifications or information you might need from our end.
> Thank you once again for your valuable time and consideration. We look forward to your response.
>
> Best regards, authors

---

### Meta-Review · Area_Chair_wbYH · 2023-12-05

**Metareview:**

The paper derives a policy gradient approach to train Generative Flow Networks and show that it outperforms existing GFlowNet training strategies.

There was a consensus among the reviewers that the paper should be rejected. The key reason is the lack of convincing experimental results, both in terms of baselines and ablations. Furthermore, the connection to RL could be improved.

**Justification For Why Not Higher Score:**

There was a consensus among the reviewers that the paper should be rejected. The key reason is the lack of convincing experimental results, both in terms of baselines and ablations. Furthermore, the connection to RL could be improved.

**Justification For Why Not Lower Score:**

N/A

---

### Decision · Program_Chairs · 2024-01-16

Reject